# On the Relationship Between CoCoA and ADMM for Distributed Empirical Risk Minimization

**Runxiong Wu**  *calvin.wu@wisc.edu*
*Department of Industrial & Systems Engineering*
*University of Wisconsin–Madison*

**Andi Wang**  *andi.wang@wisc.edu*
*Department of Industrial & Systems Engineering*
*University of Wisconsin–Madison*

**Reviewed on OpenReview:** *https://openreview.net/forum?id=kLhxBDa2yD*

## Abstract

Distributed empirical risk minimization (ERM) is often studied through two influential yet seemingly separate families of methods: CoCoA-type algorithms, derived from distributed dual coordinate ascent, and ADMM-type algorithms, derived from consensus and proximal splitting. In this paper, we investigate the connection of the two types of algorithms from a unified primal-dual perspective. We show that consensus ADMM, linearized consensus ADMM, two distributed proximal ADMM variants, and ridge-regularized CoCoA can all be written in a common update form involving a global primal variable and block dual variables. This reformulation makes several previously hidden connections explicit: For ridge-regularized ERM, CoCoA coincides with a particular proximal ADMM scheme at the level of the dual update. Moreover, consensus ADMM on the primal problem is equivalent to proximal ADMM on the dual problem under an explicit parameter mapping together with a sign reversal of the saddle objective; similar correspondences also hold for the linearized variants. These results indicates that the ADMM-type algorithms, when fine tuned, performs at least as good as CoCoA, under ridge regularized ERM problems. The unified view also yields a natural primal-dual gap stopping criterion for consensus ADMM and a unified $O(1/T)$ ergodic convergence analysis for the ADMM-type methods. Experiments on synthetic regression problems and real SVM datasets support the predicted relationships, clarify the role of tuning parameters, and show that suitably tuned ADMM variants can outperform CoCoA in the ridge-regularized setting.

## 1 Introduction

Distributed empirical risk minimization (ERM) is a central problem in modern machine learning. In this setting, $K$ machines collaboratively solve a regularized learning problem using $n$ samples $\{x_i\}_{i=1}^n \subset \mathbb{R}^d$ that are partitioned across the machines. We consider the primal problem

$$\min_{w \in \mathbb{R}^d} \mathcal{P}(w) := \frac{1}{n} \sum_{k=1}^K \sum_{i \in \mathcal{P}_k} \ell_i(w^\top x_i) + g(w), \tag{P}$$

where $\{\mathcal{P}_k\}_{k=1}^K$ denotes a partition of the dataset, $w \in \mathbb{R}^d$ is the global model, each $\ell_i$ is a convex loss function, and $g$ is a convex regularizer. Its Fenchel dual is

$$\max_{v \in \mathbb{R}^n} \mathcal{D}(v) := -\frac{1}{n} \sum_{k=1}^K \sum_{i \in \mathcal{P}_k} \ell_i^*(v_i) - g^*\left(-\frac{1}{n} X v\right), \tag{D}$$

where $v \in \mathbb{R}^n$ is the dual variable, and $\ell_i^*$ and $g^*$ denote the Fenchel conjugates of $\ell_i$ and $g$, respectively. The class of problems represented by equation P and equation D forms a foundational framework in statistical machine learning (Vapnik, 1991). Common choices for loss function $\ell_i(\cdot)$ include the squared loss, least absolute deviation, quantile loss (Koenker & Bassett Jr, 1978), Huber loss (Huber, 1992), and hinge loss for SVMs (Vapnik, 1995), while popular regularizers $g(\cdot)$ include the $\ell_1$ norm (Tibshirani, 1996), $\ell_2$ norm, and elastic net (Zou & Hastie, 2005).

A large literature has developed distributed methods for solving (equation P) and (equation D). One approach is the communication-efficient distributed dual coordinate ascent (CoCoA) family and its variants (Yang, 2013; Jaggi et al., 2014; Ma et al., 2015; Smith et al., 2015; 2018; Ma et al., 2021; Dünner et al., 2018; Lee & Chang, 2020; He et al., 2018). These methods construct local dual subproblems that can be solved in parallel across machines using coordinate ascent algorithm. Another approach is the ADMM family, including consensus ADMM, proximal ADMM, linearized ADMM, and more recent ADMM-based methods for federated learning (Boyd et al., 2011; Lin et al., 2011; Deng & Yin, 2016; Deng et al., 2017; Zhou & Li, 2023). We also refer to (Glowinski, 2014; Han, 2022; Yang et al., 2022; Maneesha & Swarup, 2021) for broader reviews of ADMM variants and applications.

In this paper, we will show an interesting connection between CoCoA with ridge regularization, consensus ADMM, and distributed proximal ADMM through a primal-dual reformulation. The technique is to reformulate these algorithms into iterative primal and block-dual update, and then unify them as proximal ADMM with a pair of distance matrices. Though the expressiveness of proximal ADMM has been revealed where a few algorithms are unified with similar technique (see Section 1.1 of Deng & Yin (2016)), the algorithms involved in our paper were not covered. Moreover, their unification is not obvious, given that the parallelization are based on distinct paradigms: CoCoA on separable dual coordinate ascent, consensus ADMM on a constrained primal reformulation with duplicated variables, and proximal ADMM

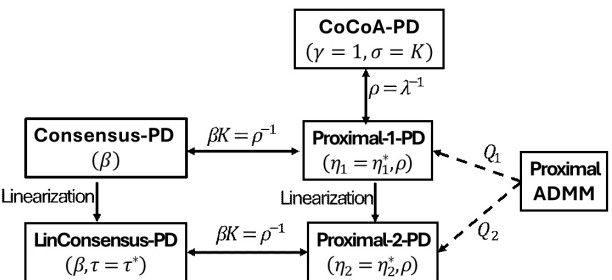

Figure 1: Connections among distributed algorithms: under $\ell_2$-regularized ERM, CoCoA is equivalent to first Proximal ADMM with $\rho = \lambda^{-1}$ on the dual variables update, and (Linearized) Consensus ADMM is equivalent to (Linearized) first Proximal ADMM when $\beta K = \rho^{-1}$.

originated in a penalized two-block ADMM format. Besides, the discovery is original from the extensive existing studies on the unification between primal-dual algorithms (Esser et al., 2010; Shefi & Teboulle, 2014; Beck, 2017) in the past few decades, which focused on associating *two-block* ADMM and its proximal variant with other algorithms such as Primal-Dual Hybrid Gradient (PDHG), Proximal Gradient Descent, Alternating Minimization, and Douglas-Rachford Splitting.

Our main result on the relationship is summarized in Figure 1. Specifically, the consensus ADMM, linearized consensus ADMM, two distributed proximal ADMM variants, and ridge-regularized CoCoA are all rewritten in a common update form. This reformulation reveals the following relations: (1) For $\ell_2$-regularized ERM, the dual update of CoCoA coincides with that of a proximal ADMM method under the parameter choice $\rho = \lambda^{-1}$. (2) The consensus ADMM applied to the primal ERM problem is equivalent to the proximal ADMM applied to the dual problem under the parameter mapping $\beta K = \rho^{-1}$. The same correspondence extends to the linearized variants. Beside, the formulation enables a unified $O(1/T)$ ergodic convergence analysis for the ADMM-type methods, including inexact subproblem updates.

This connection also provides several practical consequences. First, it is more appropriate to use proximal/consensus ADMM method than CoCoA in convex federated learning problem with ridge penalty, because it allows for the choice of tuning parameters in a wider range. Second, the reformulation yields a natural primal-dual gap, which enables developing stopping criterion for consensus ADMM. The equivalence of the methods under corresponding tuning parameters, and the fact that well-tuned ADMM variants outperform CoCoA in the ridge-regularized setting are validated by experiments on synthetic regression problems and a real SVM tasks.

The remainder of the paper is organized as follows. Section 2 reviews preliminaries on distributed primal-dual optimization. Section 3 introduces the five algorithms and casts them into the unified primal-dual form. Section 4 presents the structural connections among the methods. Section 5 gives the unified convergence analysis. Section 6 reports numerical experiments. Section 7 concludes the paper. Technical proofs are deferred to the appendix.

## 2 Preliminaries

**Notations.** Let $X = [x_1, \ldots, x_n] \in \mathbb{R}^{d \times n}$ denote the full training data matrix, where each column $x_i \in \mathbb{R}^d$ is a feature vector. The corresponding dual variable is represented by a vector $v = [v_1, \ldots, v_n]^\top \in \mathbb{R}^n$. In a distributed setting with $K$ machines, we denote by $v_{[k]} \in \mathbb{R}^{n_k}$ and $X_{[k]} \in \mathbb{R}^{d \times n_k}$ the local dual variable block and local data matrix stored on the $k$-th machine, respectively. The global data and dual variable can be expressed as block concatenations:

$$X = [X_{[1]}, \ldots, X_{[K]}], v = \left[ v_{[1]}^\top, \ldots, v_{[K]}^\top \right]^\top.$$

We define the local Fenchel conjugate loss as $\ell_{[k]}^*(v_{[k]}) := \sum_{i \in \mathcal{P}_k} \ell_i^*(v_i)$, where $\mathcal{P}_k$ is the index set of samples on machine $k$. For a symmetric positive semidefinite matrix $S$, the weighted norm is denoted by $\|x\|_S := \sqrt{x^\top S x}$, and $\lambda_{\max}(M)$ represents the largest eigenvalue of matrix $M$.

**Proximal Operator and Moreau Identity.** For a convex function $f : \mathbb{R}^m \to \mathbb{R}$ and a scalar $\lambda > 0$, the proximal operator is defined as:

$$\text{prox}_{\lambda f}(v) := \arg \min_{x \in \mathbb{R}^m} \left( f(x) + \frac{1}{2\lambda} \|x - v\|^2 \right).$$

An important identity that we use throughout the paper is the Moreau decomposition:

$$\text{prox}_{\lambda f}(v) + \lambda \, \text{prox}_{f^*/\lambda}(v/\lambda) = v,$$

where $f^*$ is the Fenchel conjugate of $f$. This identity implies that if the proximal operator of $f^*$ is computationally tractable, then the proximal operator of $f$ can be efficiently computed as well.

**Saddle-Point Reformulation.** The general ERM problem can be equivalently expressed as a saddle-point problem:

$$\min_{w \in \mathbb{R}^d} \max_{v \in \mathbb{R}^n} \left\{ L(w; v) := -\frac{1}{n} \sum_{i=1}^n \ell_i^*(v_i) + \frac{1}{n} \langle w, Xv \rangle + g(w) \right\}. \tag{SP}$$

Note that $\mathcal{D}(v) := \min_w L(w; v)$ and $\mathcal{P}(w) := \max_v L(w; v)$, we have the standard primal-dual property:

$$\mathcal{D}(v) \leq L(w; v) \leq \mathcal{P}(w), \quad \text{and} \quad \mathcal{D}(v^*) = L(w^*; v^*) = \mathcal{P}(w^*).$$

This relation ensures that the saddle-point value characterizes both the optimal primal and dual solutions. We use the primal–dual certificate to monitor convergence:

$$\text{Gap} = \mathcal{P}(w^{(t)}) - \mathcal{D}(v^{(t)}),$$

which measures the optimality gap between the primal and dual iterates $w^{(t)}$ and $v^{(t)}$ at round $t$. A smaller gap indicates that the iterates are closer to saddle-point optimality.

## 3 Distributed Algorithms via Primal and Dual Updates

In this section, we demonstrate that a variety of distributed algorithms—including the CoCoA algorithm with ridge regularization (Jaggi et al., 2014; Ma et al., 2015; 2021; Smith et al., 2018), the global consensus ADMM algorithm (Boyd et al., 2011) and its linearized variant (Lin et al., 2011), as well as two proximal ADMM methods (Deng & Yin, 2016)—can all be cast into a unified update framework involving only the primal and dual variables. As we will show in the following section, this unified formulation reveals important structural connections among these different techniques.

### 3.1 Global Consensus ADMM with Regularization

Consensus ADMM with regularization reformulates the original problem (P) into the equivalent form:

$$\min_{w \in \mathbb{R}^d} \frac{1}{n} \sum_{k=1}^{K} \sum_{i \in \mathcal{P}_k} \ell_i(w_k^\top x_i) + g(w) \quad \text{s.t.} \quad w_k = w, \ \forall k \in [K],$$

and solves it in a distributed fashion using the standard ADMM scheme (see Section 7.1.1 of Boyd et al. (2011)):

$$w_k^{(t+1)} = \arg \min_{w_k \in \mathbb{R}^d} \frac{1}{n} \sum_{i \in \mathcal{P}_k} \ell_i(w_k^\top x_i) - \langle u_k^{(t)}, w_k - w^{(t)} \rangle + \frac{\beta}{2} \| w_k - w^{(t)} \|^2, \quad \forall k \in [K],$$

$$u_k^{(t+1)} = u_k^{(t)} - \beta(w_k^{(t+1)} - w^{(t)}), \quad \forall k \in [K],$$

$$w^{(t+1)} = \arg \min_{w \in \mathbb{R}^d} g(w) - \sum_{k=1}^{K} \langle u_k^{(t+1)}, w_k^{(t+1)} - w \rangle + \sum_{k=1}^{K} \frac{\beta}{2} \| w_k^{(t+1)} - w \|^2.$$

Here, $\beta > 0$ denotes the augmented Lagrangian parameter. This algorithm can be cast into an iterative update rule **Consensus-PD** of the primal variable $w$ and the dual variable $v$, summarized in Proposition 1.

**Proposition 1** *The consensus ADMM with regularization for solving the primal problem (P) is equivalent to the following update rule:*

$$w^{(t)} = \text{prox}_{(\beta K)^{-1} g} \left( w^{(t-1)} - \frac{1}{n \beta K} X \left( 2v^{(t)} - v^{(t-1)} \right) \right), \tag{1}$$

$$v_{[k]}^{(t+1)} = \arg \min_{v_{[k]} \in \mathbb{R}^{n_k}} \frac{1}{n} \sum_{i \in \mathcal{P}_k} \ell_i^*(v_i) + \frac{1}{2n^2 \beta} \left\| v_{[k]} - v_{[k]}^{(t)} \right\|_{X_{[k]}^\top X_{[k]}}^2 - \frac{1}{n} \left\langle X_{[k]}^\top w^{(t)}, v_{[k]} \right\rangle, \quad k \in [K].$$

To simplify the updates in $v$-steps, the linearized ADMM approach (Lin et al., 2011) can be employed, resulting in **LinConsensus-PD** update rule:

$$w^{(t)} = \text{prox}_{(\beta K)^{-1} g} \left( w^{(t-1)} - \frac{1}{n \beta K} X \left( 2v^{(t)} - v^{(t-1)} \right) \right),$$

$$v_{[k]}^{(t+1)} = \text{prox}_{(n \beta / \tau) \ell_{[k]}^*} \left( v_{[k]}^{(t)} + \frac{n \beta}{\tau} X_{[k]}^\top w^{(t)} \right), \quad k \in [K], \tag{2}$$

where $\tau$ is chosen such that $\tau I_k \succeq X_{[k]}^\top X_{[k]}$ for all $k \in [K]$. The selection $\tau = \tau^* := \max \left\{ \lambda_{\max} \left( X_{[1]}^\top X_{[1]} \right), \dots, \lambda_{\max} \left( X_{[K]}^\top X_{[K]} \right) \right\}$ thereby achieves nearly optimal convergence speed.

### 3.2 Distributed Proximal ADMM

The work (Deng & Yin, 2016) introduces an additional proximal term into the standard ADMM algorithm for solving $\min_{x,y} f(x) + g(y)$ subject to $Ax + By = b$, which is referred to as generalized ADMM or proximal ADMM. Applying Algorithm 2 of the work (Deng & Yin, 2016), the proximal ADMM solving the dual problem (D) updates as follows:

$$v^{(t+1)} = \arg \min_{v \in \mathbb{R}^n} \frac{1}{n} \sum_{k=1}^{K} \sum_{i \in \mathcal{P}_k} \ell_i^*(v_i) - \langle w^{(t)}, \frac{1}{n} X v + u^{(t)} \rangle + \frac{\rho}{2} \left\| \frac{1}{n} X v + u^{(t)} \right\|^2 + \frac{1}{2} \| v - v^{(t)} \|_Q^2,$$

$$u^{(t+1)} = \arg \min_{u \in \mathbb{R}^d} g^*(u) - \langle w^{(t)}, \frac{1}{n} X v^{(t+1)} + u \rangle + \frac{\rho}{2} \left\| \frac{1}{n} X v^{(t+1)} + u \right\|^2,$$

$$w^{(t+1)} = w^{(t)} - \rho \left( \frac{1}{n} X v^{(t+1)} + u^{(t+1)} \right),$$

where $\rho > 0$ is the tuning parameter and $Q$ is a positive semi-definite matrix. To enable parallel updates of $v$ across $K$ machines, the following proposition gives two positive semi-definite matrices $Q$ choices.

**Proposition 2** *After eliminating the auxiliary variable u, the proximal ADMM updates can be simplified by choosing appropriate proximal matrices. In particular:*
*1. when $Q_1 = \frac{\rho}{n^2}(\eta_1\, diag(X_{[1]}^\top X_{[1]}, \ldots, X_{[K]}^\top X_{[K]}) - X^\top X)$, the proximal ADMM simplifies to*

$$w^{(t)} = \text{prox}_{\rho g}\left(w^{(t-1)} - \frac{\rho}{n}Xv^{(t)}\right),$$

$$v_{[k]}^{(t+1)} = \arg\min_{v_{[k]}\in\mathbb{R}^{n_k}} \frac{1}{n}\sum_{i\in\mathcal{P}_k}\ell_i^*(v_i) + \frac{\rho\eta_1}{2n^2}\left\|v_{[k]} - v_{[k]}^{(t)}\right\|_{X_{[k]}^\top X_{[k]}}^2 - \frac{1}{n}\left\langle X_{[k]}^\top\left(2w^{(t)} - w^{(t-1)}\right), v_{[k]}\right\rangle, k\in[K], \tag{3}$$

*2. when $Q_2 = \frac{\rho}{n^2}\left(\eta_2 I - X^\top X\right)$, the proximal ADMM simplifies to:*

$$w^{(t)} = \text{prox}_{\rho g}\left(w^{(t-1)} - \frac{\rho}{n}Xv^{(t)}\right),$$

$$v_{[k]}^{(t+1)} = \text{prox}_{(n/\rho\eta_2)\ell_{[k]}^*}\left(v_{[k]}^{(t)} + \frac{n}{\rho\eta_2}X_{[k]}^\top\left(2w^{(t)} - w^{(t-1)}\right)\right), k\in[K]. \tag{4}$$

The update rule of equation 3 and equation 4 are named **Proximal-1-PD** and **Proximal-2-PD**, respectively. The distributed proximal ADMM algorithms of either $Q$ are guaranteed to converge for any $\rho > 0$. The following lemma provides a reliable choice for selecting the tuning parameters $\eta_1$ and $\eta_2$ that ensures $Q_1$ and $Q_2$ to be positive semidefinite, satisfying the requirement of Deng & Yin (2016).

**Lemma 1** *For any data matrix $X$,*

$$K\, \text{diag}\left(X_{[1]}^\top X_{[1]}, \ldots, X_{[K]}^\top X_{[K]}\right) \succeq X^\top X,$$

*and thus when $\eta_1 = \eta_1^* := K, \eta_2 = \eta_2^* := K\tau^*$, $Q_1$ and $Q_2$ are positive semi-definite.*

The minimal $\eta_2$ to let $Q_2 \succeq 0$ is $\lambda_{\max}\left(X^\top X\right)$. However, this choice is practically infeasible in a distributed learning setup, as it requires the aggregation of samples from all machines.

### 3.3 CoCoA with Ridge Penalty

Unlike the aforementioned methods, which are applicable to general regularized ERM problems, the CoCoA framework was originally proposed to solve the dual of the $\ell_2$-regularized problem. Specifically, when the regularization term is the ridge penalty $g(w) = \frac{\lambda}{2}\|w\|_2^2$, CoCoA performs the following updates:

$$\tilde{v}^{(t)} = \arg\min_{v\in\mathbb{R}^n} \frac{1}{n}\sum_{k=1}^K\sum_{i\in\mathcal{P}_k}\ell_i^*(v_i) + \frac{1}{n^2\lambda}\left\langle X^\top Xv^{(t)}, v\right\rangle + \frac{\sigma}{2n^2\lambda}\sum_{k=1}^K\left\|v_{[k]} - v_{[k]}^{(t)}\right\|_{X_{[k]}^\top X_{[k]}}^2,$$

$$v^{(t+1)} = v^{(t)} + \gamma\left(\tilde{v}^{(t)} - v^{(t)}\right), \quad w^{(t+1)} = -\frac{1}{n\lambda}\sum_{k=1}^K X_{[k]}v_{[k]}^{(t+1)},$$

where the $w$-step recovers the primal variable from the dual via the KKT conditions, and the $v$-step aims to reduce the dual objective $\mathcal{D}(v)$. The parameters $\sigma$ and $\gamma$ control the approximation quality of the dual subproblem and the update aggressiveness, respectively. It has been shown in Smith et al. (2018) that setting $\gamma = 1$ and $\sigma = K$ yields the fastest guaranteed convergence.

Under these parameter choices, CoCoA simplifies to the following updates involving iterative primal and dual variables $w$ and $v$, which we refer to as **CoCoA-PD**:

$$w^{(t)} = -\frac{1}{n\lambda}\sum_{k=1}^K X_{[k]}v_{[k]}^{(t)}, \tag{5}$$

$$v_{[k]}^{(t+1)} = \arg\min_{v_{[k]}\in\mathbb{R}^{n_k}} \frac{1}{n}\sum_{i\in\mathcal{P}_k}\ell_i^*(v_i) - \frac{1}{n}\left\langle X_{[k]}^\top w^{(t)}, v_{[k]}\right\rangle + \frac{K}{2n^2\lambda}\left\|v_{[k]} - v_{[k]}^{(t)}\right\|_{X_{[k]}^\top X_{[k]}}^2, \quad k\in[K].$$

### 3.4 Summary

The five algorithms—Consensus-PD, LinConsensus-PD, Proximal-1-PD, Proximal-2-PD, and CoCoA-PD (Equations 1–5)—can all be cast into a unified primal-dual update framework. This unified view allows us to analyze the structural connections among the algorithms and to develop a common convergence analysis, as presented in Section 5.

In all algorithms, the update of the dual block $v_{[k]}$ involves applying the proximal operator $\text{prox}_{\ell_{[k]}^*}(\cdot)$ or solving a regularized quadratic problem on a linear combination of the current primal variable $w^{(t)}$ and the previous dual variable $v_{[k]}^{(t)}$. Similarly, the update of the primal variable $w$ involves a linear combination of the current iterate $w^{(t)}$, the current messages $X_{[k]}v_{[k]}^{(t+1)}$ received from individual machines, and the previous messages $X_{[k]}v_{[k]}^{(t)}$.

An immediate advantage of using the unified primal-dual update formulation, rather than the original algorithm-specific forms, is that it enables efficient evaluation of the duality gap, which provides a bound on the objective error. The duality gap can be computed by substituting the current iterates $\{v_{[k]}^{(t)}\}_{k=1}^K$ and $w^{(t)}$ into the primal objective equation P and the dual objective equation D, respectively.

**Effects of Tuning Parameters.** We summarize the selection of tuning parameters in the five algorithms mentioned above. With fixed optimal parameters $\sigma = K$ and $\gamma = 1$ in the CoCoA algorithm (Smith et al., 2018), CoCoA-PD does not have tuning parameters. The optimal selection of parameters $\eta_1, \eta_2$ in Proximal-1-PD, Proximal-2-PD, and $\tau$ in LinConsensus-PD are given in this article and confirmed by the experiments. The step sizes $\beta$ of Consensus-PD and LinConsensus-PD, and the step size $\rho$ of the Proximal-1-PD and Proximal-2-PD significantly affect the convergence speed (Boyd et al., 2011) and should be tuned in a case-specific manner, as validated in our experiments (See Section 6).

## 4 Connections Among Existing Algorithms

We now present the relationship between the algorithms from their update forms, which is described in Figure 1.

**CoCoA-PD and Proximal-1-PD.** Through the updating formula, we identified an interesting connection between CoCoA-PD and Proximal-1-PD when $g(w) = \frac{\lambda}{2}\|w\|^2$: the following corollary shows when the tuning parameters satisfies $\rho = \lambda^{-1}, \eta_1 = \sigma$, and when the CoCoA-PD selects the recommended parameter $\gamma = 1$, Proximal-1-PD and the CoCoA-PD will have identical values of dual variable updates. The result is obtained by noting that plugging the update of the primal variable $w$ into the update formula of the dual variable $v_{[k]}$ will result in the same update formula for $v_{[k]}$.

**Corollary 1 (Equivalence of CoCoA-PD and Proximal-1-PD)** *For $\ell_2$-regularized ERM problems with $g(w) = \lambda\|w\|_2^2$, the update rules in equation 3 and equation 5 produce identical dual iterates $v^{(t)}$ when $\rho = \lambda^{-1}$ and the algorithms are initialized identically.*

It is worth noting that the $w$-steps of CoCoA-PD and that of Proximal-1-PD with $g = \lambda\|w\|_2^2/2$ are different. The $w$-update for Proximal-1-PD can be represented by

$$w^{(t+1)} = \frac{1}{2}(w^{(t)} + \tilde{w}^{(t+1)}),$$

where $\{\tilde{w}(t)\}$ is the $w$-updates of CoCoA-PD. It indicates that the $w$-update of Proximal-1-PD is an exponentially weighted average of the $w$-updates of CoCoA-PD.

It is also interesting to see if the connection between CoCoA-PD and Prixmal-1-PD can be extended to other CoCoA variants with general penalty $g$ (Smith et al., 2018). To this question, we give a negative answer, because the connection in Corollary 1 relies on the same quadratic structure of the ridge penalty in CoCoA and the augmented Lagrangian in the Proximal ADMM algorithm.

The important insight from the comparison between CoCoA-PD and Proximal-1-PD is that the CoCoA-PD with the optimal selection of $\gamma$ and $\sigma$ has the same convergence rate as Proximal-1-PD, if a specific step size $\rho = \lambda^{-1}$ is selected. However, such selection may not necessarily be the optimal one that ensures fastest convergence of Proximal-1-PD. By tuning $\rho$ of Proximal-1-PD on a case-specific basis, Proximal-1-PD is able to achieve a higher convergence rate than the CoCoA-PD, as validated in our experiments.

**Consensus ADMM and Proximal ADMM.** For Consensus-PD and Proximal-1-PD, observe that the saddle-point formulation satisfies

$$\min_{w \in \mathbb{R}^d} \max_{v \in \mathbb{R}^n} L(w; v) = \max_{w \in \mathbb{R}^d} \min_{v \in \mathbb{R}^n} (-L(w; v)).$$

Hence, Proximal-1-PD can be interpreted as applying Consensus-PD to the equivalent saddle-point problem with negated objective $-L(w; v)$. This equivalence is formalized in the following corollary.

**Corollary 2 (Equivalence of Consensus ADMM and Proximal ADMM)** *Assume identical initialization and augmented Lagrangian parameters satisfying $\beta K = \rho^{-1}$ in Consensus ADMM and Proximal ADMM. Then, we have (1) Consensus-PD is equivalent to Proximal-1-PD when $\eta_1 = K$, (2) LinConsensus-PD is equivalent to Proximal-2-PD when $\eta_2 = K\tau$.*

Combining Corollaries 1 and 2, we observe that CoCoA variants arise as special cases of consensus ADMM under specific parameter settings, challenging the conclusion in Smith et al. (2018) that CoCoA is fundamentally distinct.

## 5 Theoretical Analysis

Representing the four ADMM-based primal-dual update forms 1–4 into the generic update rules also provides a unified and straightforward approach of their convergence analysis. Using the convergence analysis framework of (He & Yuan, 2012; Lu & Yang, 2023), we establish an $O(1/T)$ ergodic rates of these algorithms. To proceed, we first show that each algorithm can be viewed as Lu & Yang (2023) applied to the Lagrangian saddle-point problem, as formalized below.

**Lemma 2** *Let $z = (w, v)$ denote the concatenated primal and dual variables, and define the monotone operator*

$$\mathcal{F}(z) = \begin{pmatrix} \partial_w L(w, v) \\ -\partial_v L(w, v) \end{pmatrix},$$

*where $L(w, v)$ is the Lagrangian of the saddle-point problem. Then, the update rules of the algorithms (1), (2), (3), and (4) can all be written in the generic proximal form:*

$$P(z^{(t)} - z^{(t+1)}) \in \mathcal{F}(z^{(t+1)}),$$

*with the corresponding matrix $P$ specified as follows:*

$$P_1 = \begin{pmatrix} \beta K I & \frac{1}{n}A \\ \frac{1}{n}A^\top & \frac{1}{n^2\beta}B \end{pmatrix}, P_2 = \begin{pmatrix} \beta K I & \frac{1}{n}A \\ \frac{1}{n}A^\top & \frac{\tau}{n^2\beta}I \end{pmatrix}, P_3 = \begin{pmatrix} \rho^{-1}I & -\frac{1}{n}A \\ -\frac{1}{n}A^\top & \frac{\rho\eta_1}{n^2}B \end{pmatrix}, P_4 = \begin{pmatrix} \rho^{-1}I & -\frac{1}{n}A \\ -\frac{1}{n}A^\top & \frac{\rho\eta_2}{n^2}I \end{pmatrix}$$

*where $A = X$, and $B = \mathrm{diag}\left(X_{[1]}^\top X_{[1]}, \ldots, X_{[K]}^\top X_{[K]}\right)$.*

As the algorithm-specific matrices $P_1, \ldots, P_4$ are positive semidefinite, the convergence analysis can be conducted within the standard framework of generalized PPMs (Lu & Yang, 2023; Lu et al., 2017). Specifically, Theorem 1 characterizes the convergence behavior of the general distributed primal–dual algorithmic framework.

**Theorem 1** *Let $\left\{z^{(t)} = (w^{(t)}, v^{(t)})\right\}_{t=0}^\infty$ be the sequence generated by the generic update rule in Lemma 2 with a positive definite matrix $P$ and the initial point $z^{(0)} = (w^{(0)}, v^{(0)})$. For any $z = (w, v)$, the following inequality holds:*

$$L(\bar{w}^{(T)}; v) - L(w; \bar{v}^{(T)}) \leq \frac{\|z - z^{(0)}\|_P^2}{2T},$$

*where $\bar{z}^{(T)} = (\bar{w}^{(T)}, \bar{v}^{(T)}) = \frac{1}{T}\sum_{t=1}^{T} z^{(t)}$.*

By selecting $z = (w^*, v^*)$, the optimal solutions to equation P and equation D, Theorem 1 indicates that $L(\bar{w}^{(T)}; v^*) - L(w^*; \bar{v}^{(T)})$ converges to zero. Under the assumption that objective being strongly convex-concave or that the saddle point being unique, $\bar{z}^{(T)}$ converges to the optimal solution at a rate of $O(1/T)$.

In practice, all $v$-steps of the updates 1–4 solve a minimization problem which would rely on an inner loop, in case no closed-form solution is available. We present a unified proof of convergence for 1–4 which addresses the inexact updates, based on the technique of Lu & Yang (2023).

**Theorem 2** *Let $P$ be positive definite, and $z^* = (w^*, v^*)$ be the optimal solution of the saddle point problem (SP). If the sequence $\{z^{(t)}\}$ satisfies $P(z^{(t)} - z^{(t+1)}) + \epsilon^{(t+1)} \in \mathcal{F}(z^{(t+1)})$ with $\sum_{t=1}^{\infty} \|\epsilon^{(t)}\|_2 < \infty$, then there exists a constant $D < \infty$ such that $\sup_t \|z^* - z^{(t)}\| \le D$ and*

$$L(\bar{w}^{(T)}; v^*) - L(w^*; \bar{v}^{(T)}) \le \frac{\|z^* - z^{(0)}\|_P^2}{2T} + \frac{D\sum_{t=1}^{T}\|\epsilon^{(t)}\|_2}{T}.$$

In this theorem, $\{z^{(t)}\}$ is the sequence generated by the inexact algorithm subject to inner-loop computational errors, $\epsilon^{(t)}$ represents the computational error incurred due to the inexact update of iteration $t$. Under the assumption that the total error over all iterations is bounded, an $O(1/T)$ convergence rate can be achieved.

One approach of specifying the number of iterations is to ensure $\|\epsilon^{(t)}\|$ is below a limit, for example $1/t^2$. If a standard inner solver such as Gradient Descent exhibits linear convergence: $\|\epsilon^{(t,j)}\|_2 \le C \cdot \rho^j$, the number of inner iterations $j$ must be increased logarithmically. However, for some losses such as logistic loss, the conjugate is not globally smooth, higher number of inner iterations may be needed. In practice, however, it is unnecessary to increase the number of inner iterations indefinitely, as the sub-gradient error eventually reaches the machine precision. At this stage, further inner iterations do not yield meaningful improvements.

**Remark 1** *Theorem 1 and Theorem 2 require the stronger assumption $P \succ 0$. For the four matrices considered above, strict positive definiteness is guaranteed under the following sufficient and necessary conditions specified by the Schur Complement Theorem:*

$$P_1: \ \beta > 0, \ \text{each } X_{[k]} \text{ has full column rank, and } K^{-1}B \succ A^\top A,$$
$$P_2: \ \beta > 0, \ K\tau I \succ A^\top A,$$
$$P_3: \ \rho > 0, \ \text{each } X_{[k]} \text{ has full column rank, and } \eta_1 B \succ A^\top A,$$
$$P_4: \ \rho > 0, \ \eta_2 I \succ A^\top A.$$

*In particular, for sufficiently large $K\tau$ or sufficiently large $\eta_2$, $P_2$ and $P_4$ are positive definite, respectively. If some block $X_{[k]}$ is rank deficient, then $B = \text{diag}(X_{[1]}^\top X_{[1]}, \ldots, X_{[K]}^\top X_{[K]})$ is singular, so $P_1$ and $P_3$ are not strictly positive definite, thus the convergence results have to be established in alternative approaches. Alternatively, as long as all blocks $X_{[k]}$ are not rank deficient, for sufficiently small $K$ or sufficiently large $\eta_1$, $P_1$ and $P_3$ are positive definite.*

## 6 Experiments

In this section, we perform experiments to test the performance of the primal-dual update rules under different parameter settings. Specifically, we first studying how the the tuning parameters affect each algorithm, to verify our suggestions on tuning parameter selection. Then, we evaluate the performance of the five update rules using synthetic data for Lasso and Ridge regression tasks, which also verify the equivalency results in Section 4. Additional evaluations on the performance of the five update rules on three real-world binary classification tasks employing SVM are included in Appendix D.2. All experiments are conducted on the Dell Latitude 7450 Laptop, and each can be finished within 6 hours. The codes are available in this Google Drive folder.

**Experiment 1.** We aim to verify the effect of $\eta_1, \eta_2, \tau$ for the Proximal-1-PD, Proximal-2-PD, and LinConsensus-PD update rules, and compare them with the effect of the step sizes $\rho$ and $\beta$. We used

`a1a` dataset in the LibSVM library (Chang & Lin, 2011): `a1a`, `w8a`, and `real-sim`. Details of this dataset, including the number of samples, features, and clients, are included in Table 1. In the problem, we evenly distribute the data into $K = 10$ machines. We train the model using $\ell_2$-regularized SVM model with regularization parameter of $\lambda = 1/n$. We tested the performance of three algorithms, where Proximal-1-PD is subject to the tuning parameters $\eta_1$ and $\rho$, Proximal-2-PD is subject to the tuning parameters $\eta_2$ and $\rho$, and LinConsensus-PD is subject to $\tau$ and $\beta$. In the experiments, we test each method through fixing one tuning parameter and setting multiple values for the other tuning parameter. We record the trajectory of the relative gap difference in 500 communication rounds.

Table 1: Description of the datasets.

| Dataset | $n$ | $d$ | $K$ |
|---|---|---|---|
| `a1a` | 1605 | 119 | 10 |
| `w8a` | 49749 | 300 | 60 |
| `real-sim` | 72309 | 20958 | 100 |

**Results.** The trajectory of the gap are shown in Figure 2. The first row demonstrated the validity of the selection of $\eta_1, \eta_2$ and $\tau$ for the three methods. The recommended values, denoted by light blue lines, ensure the convergence of the algorithm. When these values become smaller, the convergence speed increases only slightly (e.g., the green and purple line). When these values become too small, however, the algorithms may fail to converge. Second, we can see from the three figures in second row that the ADMM step size parameter $\rho$ and $\beta$ has significantly impacts the algorithms' performance.

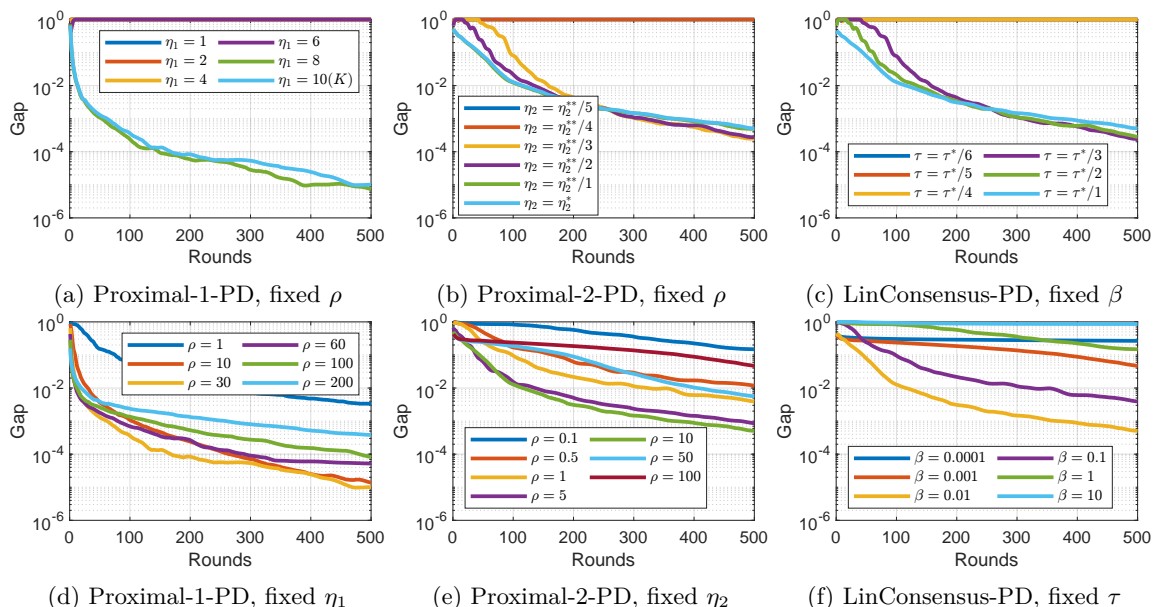

Figure 2: Effect of tuning parameters on various distributed algorithms in Experiment 1.

**Experiment 2.** We test five the update rules on Ridge Regression problem and LASSO problem, where each $\ell_i = \frac{1}{2}(y_i - x_i^\top w)^2$, using synthetic data. The data generation mechanism is detailed in Appendix D.1. We run the five update rules to solve the Ridge Regression problem on IID and non-IID dataset, and run the four ADMM algorithms to solve the LASSO problem. In these update rules, we select the suggested value of $\gamma, \sigma, \eta_1, \eta_2$, and $\tau$, and select the optimal $\beta$ or $\rho$ to achieve optimal performance. Notably, it has been observed that the optimal $\beta$ and $\rho$ in Consensus-PD and Proximal-1-PD satisfies $\beta K = \rho^{-1}$, and so are the optimal $\beta$ and $\rho$ in LinConsensus-PD and Proximal-2-PD, indicating their connection.

**Results.** We present the simulation results in Figure 3. We observe that the performance of Consensus-PD and Proximal-1-PD are almost identical, while the performance of LinConsensus-PD and Proximal-2-PD are

almost identical. These simulation results further confirm the strong connection between these two pairs. All four ADMM variants, with the optimized tuning parameters, significantly outperform the CoCoA framework. This is because of CoCoA is the Proximal-1-PD with a specific step size $\rho = \lambda^{-1}$. The figure also shows that Consensus-PD and Proximal-1-PD achieve smaller relative gap difference compared with LinConsensus-PD and Proximal-2-PD in the same amount of rounds, though the computation for the latter two variants are significantly simpler.

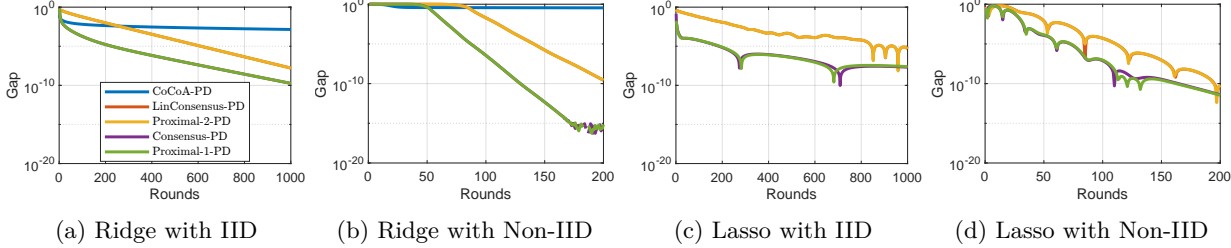

(a) Ridge with IID  (b) Ridge with Non-IID  (c) Lasso with IID  (d) Lasso with Non-IID

Figure 3: Relative gap difference versus the number of communication rounds for various synthetic datasets when using different update rules in Experiment 2.

## 7 Conclusion

In this article, we unified distributed primal-dual algorithms, including CoCoA, two proximal ADMM algorithms, consensus ADMM, and linearized ADMM into updates rule that only involve the primal and dual variable updates. Among them, the two proximal ADMM algorithms are new, obtained from choosing two positive definite matrices to enable the proximal ADMM algorithm to solve distributed, regularized federated learning problem. The unified update rules reveal that the CoCoA algorithm can be interpreted as a special case of proximal ADMM with a specific tuning parameter, and proximal ADMM and consensus ADMM are equivalent. The findings in the paper also indicated rich expressiveness of distributed learning that involves global primal updates and local dual updates. This framework enables the use of the gap between the primal and dual objectives as a stopping criterion for the consensus ADMM algorithm, and also enables us to use a simple and unified ergodic convergence analysis for ADMM variants. By thoroughly investigating the influence of tuning parameters on convergence speed, we found that all ADMM variants consistently outperform the CoCoA-PD with properly selected tuning parameters, as validated by the experiments with synthetic and real-world datasets.

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

# A   Derivation of the Dual Problem

Let $w^\top x_i = u_i$ for any $i = 1, \ldots, n$, we can equivalently transform the original problem equation P as the following form:

$$\min_{w \in \mathbb{R}^d, u \in \mathbb{R}^n} \frac{1}{n} \sum_{i=1}^{n} \ell_i(u_i) + g(w) \quad \text{s.t. } w^\top x_i = u_i, \quad i = 1, \ldots, n. \tag{6}$$

By introducing the Lagrangian multiplier $v = [v_1, \ldots, v_n]^\top$, we can write the Lagrangian function as

$$L(w, u; v) := \frac{1}{n} \sum_{i=1}^{n} \ell_i(u_i) + g(w) + \frac{1}{n} \sum_{i=1}^{n} v_i(w^\top x_i - u_i).$$

Note that we incorporate the fraction constant $\frac{1}{n}$ into the Lagrange multiplier to ensure alignment with the loss function when minimizing the Lagrangian function for the primal variables. Thus, the dual problem could be obtained by taking the infimum to both $w$ and $u$:

$$
\begin{aligned}
\inf_{w,u} L(w, u; v) &= \inf_u \left\{ \frac{1}{n} \sum_{i=1}^{n} (\ell_i(u_i) - v_i u_i) \right\} + \inf_w \left\{ g(w) + \langle w, \frac{1}{n} \sum_{i=1}^{n} v_i x_i \rangle \right\} \\
&= -\frac{1}{n} \sum_{i=1}^{n} \ell_i^*(v_i) - g^* \left( -\frac{1}{n} \sum_{i=1}^{n} v_i x_i \right).
\end{aligned}
$$

After changing the sign to make the maximization of the dual problem into the minimization, we have the following dual formulation:

$$\min_{v \in \mathbb{R}^n} \left\{ \mathcal{D}(v) := \frac{1}{n} \sum_{i=1}^{n} \ell_i^*(v_i) + g^* \left( -\frac{1}{n} \sum_{i=1}^{n} v_i x_i \right) \right\}.$$

For the distributed problem form equation D, the corresponding distributed dual problem form is thus

$$\min_{v \in \mathbb{R}^n} \left\{ \mathcal{D}(v) := \frac{1}{n} \sum_{k=1}^{K} \sum_{i \in \mathcal{P}_k} \ell_i^*(v_i) + g^* \left( -\frac{1}{n} \sum_{k=1}^{K} \sum_{i \in \mathcal{P}_k} v_i x_i \right) \right\}.$$

Furthermore, the KKT conditions are listed as follows:

$$
\begin{cases}
x_i^\top w^* = u_i^*, & i = 1, \ldots, n, \\
v_i^* \in \partial \ell_i(u_i^*), & i = 1, \ldots, n, \\
-\dfrac{1}{n} \displaystyle\sum_{i=1}^{n} v_i^* x_i \in \partial g(w^*).
\end{cases}
$$

After simplification, we have

$$
\begin{cases}
x_i^\top w^* = \operatorname{Prox}_{\ell_i}\left(x_i^\top w^* + v_i^*\right), & \text{for any } i = 1, \ldots, n, \\
w^* = \operatorname{Prox}_g\left(w^* - \dfrac{1}{n}\displaystyle\sum_{i=1}^{n} v_i^* x_i\right).
\end{cases}
$$

# B    Proofs for the Results in Section 3

## B.1   Proof of Proposition 1

To better understand the procedure of consensus ADMM, we focus on the dual form of the $w_k$-update problem for the $k$-th agent. Let $w_k^\top x_i = \tilde{u}_i$ for any $i \in \mathcal{P}_k$. Using this substitution, we can equivalently rewrite the original problem in the following form:

$$
\min_{w_k \in \mathbb{R}^d, \tilde{u}_{[k]} \in \mathbb{R}^{n_k}} \frac{1}{n} \sum_{i \in \mathcal{P}_k} \ell_i(\tilde{u}_i) + \frac{\beta}{2}\|w_k - w^{(t)} - \beta^{-1} u_k^{(t)}\|^2 \quad \text{s.t.} \quad w_k^\top x_i = \tilde{u}_i,\ i \in \mathcal{P}_k. \tag{7}
$$

By introducing the Lagrange multiplier $\tilde{v}_{[k]} \in \mathbb{R}^{n_k}$, the Lagrangian function becomes:

$$
L(w_k, \tilde{u}_{[k]}; \tilde{v}_{[k]}) := \frac{1}{n} \sum_{i \in \mathcal{P}_k} \ell_i(\tilde{u}_i) + \frac{\beta}{2}\|w_k - w^{(t)} - \beta^{-1} u_k^{(t)}\|^2 + \frac{1}{n} \sum_{i \in \mathcal{P}_k} \tilde{v}_i(w_k^\top x_i - \tilde{u}_i).
$$

Taking the infimum of the Lagrangian with respect to $w_k$ and $\tilde{u}_{[k]}$, we derive the dual form of this subproblem:

$$
\begin{aligned}
\min_{\tilde{v}_{[k]} \in \mathbb{R}^{n_k}} \ \ & \frac{1}{n} \sum_{i \in \mathcal{P}_k} \ell_i^*(\tilde{v}_i) + \frac{1}{2n^2\beta}\left(\tilde{v}_{[k]} - \tilde{v}_{[k]}^{(t)}\right)^\top X_{[k]}^\top X_{[k]} \left(\tilde{v}_{[k]} - \tilde{v}_{[k]}^{(t)}\right) \\
& - \frac{1}{n}\left\langle X_{[k]}^\top \left(w^{(t)} + \frac{1}{\beta}u_k^{(t)} - \frac{1}{n\beta}X_{[k]}\tilde{v}_{[k]}^{(t)}\right), \tilde{v}_{[k]} - \tilde{v}_{[k]}^{(t)}\right\rangle.
\end{aligned} \tag{8}
$$

Let $\tilde{v}_{[k]}^{(t+1)}$ denote the optimal solution of the above dual problem. Since $w^{(t+1)}$ is the optimal primal solution, the KKT conditions between the primal and dual solutions imply:

$$
w_k^{(t+1)} = w^{(t)} + \frac{1}{\beta}u_k^{(t)} - \frac{1}{n\beta}X_{[k]}\tilde{v}_{[k]}^{(t+1)}.
$$

Substituting the above relationship into the $u_k^{(t+1)}$ update formula, we obtain:

$$
u_k^{(t+1)} = \frac{1}{n}X_{[k]}\tilde{v}_{[k]}^{(t+1)}.
$$

We can further simplify the $w_k^{(t+1)}$ update as:

$$
w_k^{(t+1)} = w^{(t)} + \frac{1}{n\beta}X_{[k]}\left(\tilde{v}_{[k]}^{(t)} - \tilde{v}_{[k]}^{(t+1)}\right).
$$

Representing $w_k^{(t)}$ and $u_k^{(t)}$ in terms of $w^{(t)}$ and $\tilde{v}_{[k]}^{(t)}$ in the consensus ADMM updates, we derive the following updates:

For the dual variable $\tilde{v}_{[k]}$:

$$\tilde{v}_{[k]}^{(t+1)} \approx \underset{\tilde{v}_{[k]} \in \mathbb{R}^{n_k}}{\arg\min} \frac{1}{n} \sum_{i \in \mathcal{P}_k} \ell_i^*(\tilde{v}_i) + \frac{1}{2n^2\beta} \left( \tilde{v}_{[k]} - \tilde{v}_{[k]}^{(t)} \right)^\top X_{[k]}^\top X_{[k]} \left( \tilde{v}_{[k]} - \tilde{v}_{[k]}^{(t)} \right)$$

$$- \frac{1}{n} \left\langle X_{[k]}^\top w^{(t)}, \tilde{v}_{[k]} \right\rangle, \quad k \in [K] \text{ (in parallel)}.$$

For the primal variable $w^{(t+1)}$:

$$w^{(t+1)} = \text{prox}_{(\beta K)^{-1} g} \left( w^{(t)} - \frac{1}{n\beta K} X \left( 2\tilde{v}^{(t+1)} - \tilde{v}^{(t)} \right) \right).$$

To prove that the dual iterate $\tilde{v}^{(t)}$ converges to the optimal dual solution $v^*$, we invoke Lemma 2, which shows that the above iteration can be equivalently expressed as

$$P\left( z^{(t)} - z^{(t+1)} \right) = \begin{pmatrix} \beta K I & \frac{1}{n} X \\ \frac{1}{n} X^\top & \frac{1}{n^2\beta} \text{diag}\left( X_{[1]}^\top X_{[1]}, \ldots, X_{[K]}^\top X_{[K]} \right) \end{pmatrix} \begin{pmatrix} w^{(t)} - w^{(t+1)} \\ v^{(t)} - v^{(t+1)} \end{pmatrix} \in \mathcal{F}\left( z^{(t+1)} \right),$$

where $z = (w, v)$. It then follows from Theorems 1 and 2 that $\tilde{v}^{(t)}$ converges to $v^*$ and $w^{(t)}$ converges to the optimal primal solution $w^*$. Hence, the proof is complete. Note that the proofs of Theorems 1 and 2, presented in Appendix C.2 and C.3, do not rely on this Proposition. By further linearizing the local data matrix $X_{[k]}^\top X_{[k]}$ in the dual variable $v$ update, we derive the corresponding update formula for the consensus ADMM incorporating linearization techniques.

## B.2  Proof of Proposition 2

For the first matrix choice of $Q = \frac{\rho}{n^2} \left( \eta_1 \text{diag}\left( X_{[1]}^\top X_{[1]}, \ldots, X_{[K]}^\top X_{[K]} \right) - X^\top X \right)$, the proximal ADMM updates can be equivalently written as:

$$v^{(t+1)} \approx \underset{v_{[k]} \in \mathbb{R}^{n_k}}{\arg\min} \frac{1}{n} \sum_{k=1}^{K} \sum_{i \in \mathcal{P}_k} \ell_i^*(v_i) + \frac{\rho \eta_1}{2n^2} \sum_{k=1}^{K} \left( v_{[k]} - v_{[k]}^{(t)} \right)^\top X_{[k]}^\top X_{[k]} \left( v_{[k]} - v_{[k]}^{(t)} \right)$$

$$+ \left\langle \frac{\rho}{n} X^\top \left( \frac{1}{n} X v^{(t)} + u^{(t)} - \rho^{-1} w^{(t)} \right), v - v^{(t)} \right\rangle,$$

$$u^{(t+1)} = \text{Prox}_{\rho^{-1} g^*} \left( \rho^{-1} w^{(t)} - \frac{1}{n} X v^{(t+1)} \right),$$

$$w^{(t+1)} = w^{(t)} - \rho \left( \frac{1}{n} X v^{(t+1)} + u^{(t+1)} \right).$$

For the $v$-update, note that the update formula for the primal variable $w$ satisfies:

$$\frac{1}{n} X v^{(t)} + u^{(t)} - \frac{1}{\rho} w^{(t)} = \frac{1}{\rho} \left( w^{(t-1)} - 2w^{(t)} \right).$$

Substituting this relationship into the dual variable $v$-update formula and simplifying in parallel, we immediately obtain the corresponding update formula for $v$. For the $w$-update, using the Moreau identity $\text{prox}_{\lambda f}(v) + \lambda \text{prox}_{f^*/\lambda}(v/\lambda) = v$, we have:

$$w^{(t+1)} = \rho \left( \rho^{-1} w^{(t)} - \frac{1}{n} X v^{(t+1)} - u^{(t+1)} \right)$$

$$= \rho \left( \rho^{-1} w^{(t)} - \frac{1}{n} X v^{(t+1)} - \text{Prox}_{\rho^{-1} g^*} \left( \rho^{-1} w^{(t)} - \frac{1}{n} X v^{(t+1)} \right) \right)$$

$$= \text{Prox}_{\rho g} \left( w^{(t)} - \frac{\rho}{n} X v^{(t+1)} \right).$$

Thus, we can equivalently transform the proximal ADMM with the first matrix choice of $Q$ as the corresponding update formula. Further linearizing the local data matrix $X_{[k]}^\top X_{[k]}$, we can obtain the corresponding update formula for the proximal ADMM with the second matrix choice of $Q$.

### B.3 Proof of Lemma 1

For any vector $u = [u_{[1]}, \ldots, u_{[K]}]^\top \in \mathbb{R}^n$ with each $u_{[k]} \in \mathbb{R}^{n_k}$, we have:

$$
\begin{aligned}
u^\top X^\top X u &= K^2 \left\| \frac{1}{K} \sum_{k=1}^K X_{[k]} u_{[k]} \right\|^2, \\
&\leq K^2 \cdot \frac{1}{K} \sum_{k=1}^K \left\| X_{[k]} u_{[k]} \right\|^2, \\
&= K \sum_{k=1}^K \left\| X_{[k]} u_{[k]} \right\|^2, \\
&= u^\top K \operatorname{diag}\left( X_{[1]}^\top X_{[1]}, \ldots, X_{[K]}^\top X_{[K]} \right) u.
\end{aligned}
$$

The second inequality holds due to the convexity property of the squared norm, $\| \cdot \|^2$. Thus, we conclude that:

$$
K \operatorname{diag}\left( X_{[1]}^\top X_{[1]}, \ldots, X_{[K]}^\top X_{[K]} \right) \succeq X^\top X.
$$

Based on the above relationship, it is straightforward to verify that these tuning parameters

$$
\eta_1 = K \text{ and } \eta_2 = K \max\left\{ \lambda_{\max}\left( X_{[1]}^\top X_{[1]} \right), \ldots, \lambda_{\max}\left( X_{[K]}^\top X_{[K]} \right) \right\}
$$

ensure that the matrix $Q$ is positive semi-definite.

### B.4 Proof of Corollary 1

Substituting $g(w) = \frac{\lambda}{2} \|w\|^2$ into the updates of equation 3, we immediately simplifies the updates of Proximal-1-PD as follows:

$$
\begin{aligned}
v_{[k]}^{(t+1)} &= \arg\min_{v_{[k]} \in \mathbb{R}^{n_k}} \frac{1}{n} \sum_{i \in \mathcal{P}_k} \ell_i^*(v_i) + \frac{\rho \eta_1}{2n^2} \left\| v_{[k]} - v_{[k]}^{(t)} \right\|_{X_{[k]}^\top X_{[k]}}^2 - \frac{1}{n} \left\langle X_{[k]}^\top \left( 2w^{(t)} - w^{(t-1)} \right), v_{[k]} \right\rangle, k \in [K] \\
w^{(t+1)} &= \frac{1}{\lambda \rho + 1} \left( w^{(t)} - \frac{\rho}{n} X v^{(t+1)} \right).
\end{aligned}
$$

Considering $\rho = \frac{1}{\lambda}$, we have by the $w$-update

$$
w^{(t+1)} = \frac{1}{2} \left( w^{(t)} - \frac{1}{n\lambda} X v^{(t+1)} \right), \text{ for any } t.
$$

Substituting the above relationship with timestep $t$ into the $v$-update gives us

$$
v_{[k]}^{(t+1)} = \arg\min_{v_{[k]} \in \mathbb{R}^{n_k}} \frac{1}{n} \sum_{i \in \mathcal{P}_k} \ell_i^*(v_i) + \frac{\eta_1}{2n^2 \lambda} \left\| v_{[k]} - v_{[k]}^{(t)} \right\|_{X_{[k]}^\top X_{[k]}}^2 + \frac{1}{n^2 \lambda} \left\langle X_{[k]}^\top X v^{(t)}, v_{[k]} \right\rangle, k \in [K].
$$

Compared to the CoCoA-PD update, this update formula matches it when $\eta_1 = \sigma$, but differs in the $w$-update.

## B.5   Proof of Corollary 2

To see the equivalence between Consensus-PD and Proximal-1-PD algorithms, let's consider both algorithms applied to the following saddle-point formulation of the general empirical risk minimization problem:

$$\min_{w} \max_{v} L(w, v) := \frac{1}{n} \sum_{i=1}^{n} (v_i \langle w, x_i \rangle - \ell_i^*(v_i)) + g(w),$$

where the loss function $\ell_i(w^\top x_i)$ is represented in its convex conjugate form as $\ell_i(w^\top x_i) = \sup_{v_i \in \mathbb{R}} \{v_i \langle w, x_i \rangle - \ell_i^*(v_i)\}$. The Consensus-PD update (see Proposition 1 of our paper) is

$$v_{[k]}^{(t+1)} = \arg\min_{v_{[k]} \in \mathbb{R}^{n_k}} \frac{1}{n} \sum_{i \in \mathcal{P}_k} \ell_i^*(v_i) - \frac{1}{n} \left\langle X_{[k]}^\top w^{(t)}, v_{[k]} \right\rangle + \frac{1}{2n^2\beta} \left\| v_{[k]} - v_{[k]}^{(t)} \right\|_{X_{[k]}^\top X_{[k]}}^2, \quad k \in [K]$$

$$w^{(t+1)} = \mathrm{prox}_{(\beta K)^{-1} g} \left( w^{(t)} - \frac{1}{n\beta K} X \left( 2v^{(t+1)} - v^{(t)} \right) \right).$$

It can be equivalently written as

$$v^{(t+1)} = \arg\max_{v} L(w^{(t)}, v) - \frac{s_1}{2} \|v - v^{(t)}\|_{M_1}^2,$$

$$w^{(t+1)} = \arg\min_{w} L(w, 2v^{(t+1)} - v^{(t)}) + \frac{s_2}{2} \|w - w^{(t)}\|_{M_2}^2,$$

where $s_1 = \frac{1}{n^2\beta}$, $s_2 = \beta K$, $M_1 = \mathrm{diag}\left(X_{[1]}^\top X_{[1]}, \dots, X_{[K]}^\top X_{[K]}\right)$, and $M_2 = I$ are the algorithm dependent parameters.

Now, consider instead solving the problem $\max_{v} \min_{w} L(w, v)$, which is equivalent with solving $\min_{v} \max_{w} - L(w, v)$. The same algorithm Consensus-PD ($\arg\max$ first and then $\arg\min$) can be applied on this problem, leading to:

$$w^{(t+1)} = \arg\max_{w} - L(w, v^{(t)}) - \frac{s_2}{2} \|w - w^{(t)}\|_{M_2}^2,$$

$$v^{(t+1)} = \arg\min_{v} - L(2w^{(t+1)} - w^{(t)}, v) + \frac{s_1}{2} \|v - v^{(t)}\|_{M_1}^2.$$

Note that the maximization step uses the previous dual iterate $v^{(t)}$, whereas the minimization step is evaluated at the extrapolated primal point $2w^{(t+1)} - w^{(t)}$. To facilitate comparison with the Proximal-1-PD algorithm, we introduce the shifted primal sequence $\tilde{w}^{(t)} := w^{(t+1)}$. We have

$$2w^{(t+1)} - w^{(t)} = 2\tilde{w}^{(t)} - \tilde{w}^{(t-1)}.$$

Dropping the tilde for notational simplicity, the above iteration is equivalent with

$$v^{(t+1)} = \arg\min_{v} \left\{ -L(2w^{(t)} - w^{(t-1)}, v) + \frac{s_1}{2} \|v - v^{(t)}\|_{M_1}^2 \right\},$$

$$w^{(t+1)} = \arg\max_{w} \left\{ -L(w, v^{(t+1)}) - \frac{s_2}{2} \|w - w^{(t)}\|_{M_2}^2 \right\}.$$

Substituting $s_1 = \frac{1}{n^2\beta}, s_2 = \beta K, M_1 = \mathrm{diag}\left(X_{[1]}^\top X_{[1]}, \dots, X_{[K]}^\top X_{[K]}\right), M_2 = I$ into the above iteration yields

$$v_{[k]}^{(t+1)} = \arg\min_{v_{[k]} \in \mathbb{R}^{n_k}} \frac{1}{n} \sum_{i \in \mathcal{P}_k} \ell_i^*(v_i) - \frac{1}{n} \left\langle X_{[k]}^\top \left(2w^{(t)} - w^{(t-1)}\right), v_{[k]} \right\rangle + \frac{1}{2n^2\beta} \left\| v_{[k]} - v_{[k]}^{(t)} \right\|_{X_{[k]}^\top X_{[k]}}^2,$$

$$w^{(t+1)} = \mathrm{prox}_{(\beta K)^{-1} g} \left( w^{(t)} - \frac{1}{n\beta K} X v^{(t+1)} \right).$$

which is the same as the Proximal-1-PD update form (see Equation equation 3 of Proposition 2) under the parameter setting $\eta_1 = \eta^* = K$ and $\rho = \frac{1}{\beta K}$.

Because of the convex-concave structure of the saddle-point function $L(w, v)$, we know that $\min_w \max_v L(w, v)$ and $\max_v \min_w L(w, v)$ have the same solution. Therefore, in summary, Proximal-1-PD is just to use Consensus-PD to solve the max-min problem. A similar derivation holds for LinConsensus-PD and Proximal-2-PD. Thus, we verify the Corollary 2.

## C Proofs for the Results in Section 5

### C.1 Proof of Lemma 2

Notice that we have $\mathcal{F}(z^{(t+1)}) = \begin{pmatrix} \frac{1}{n} X v^{(t+1)} + \partial g(w^{(t+1)}) \\ \frac{1}{n} \partial \ell^*(v^{(t+1)}) - \frac{1}{n} X^\top w^{(t+1)} \end{pmatrix}$ for the min-max objective function defined in Section 5. Next, we would derive the corresponding semi-positive matrix $P$ individually according to the different algorithm updates.

**Distributed proximal ADMM.** For the distributed proximal ADMM with the first matrix choice, we consider the update rules by updating the primal variable $w$ first and then the dual variable $v$ as follows:

$$w^{(t+1)} = \text{prox}_{\rho g}\left(w^{(t)} - \frac{\rho}{n} X v^{(t)}\right),$$

$$v_{[k]}^{(t+1)} \approx \arg\min_{v_{[k]} \in \mathbb{R}^{n_k}} \frac{1}{n} \sum_{i \in \mathcal{P}_k} \ell_i^*(v_i) + \frac{\rho \eta_1}{2n^2}\left(v_{[k]} - v_{[k]}^{(t)}\right)^\top X_{[k]}^\top X_{[k]}\left(v_{[k]} - v_{[k]}^{(t)}\right)$$
$$- \frac{1}{n}\left\langle X_{[k]}^\top\left(2w^{(t+1)} - w^{(t)}\right), v_{[k]}\right\rangle, \quad k \in [K] \text{ (in parallel)}.$$

By utilizing the first-order optimality conditions, we can equivalently transform the above update rules as follows:

$$0 \in \partial g(w^{(t+1)}) + \rho^{-1}\left(w^{(t+1)} - w^{(t)} + \frac{\rho}{n} X v^{(t)}\right),$$

$$0 \in \frac{1}{n}\partial\ell^*(v^{(t+1)}) + \frac{\rho\eta_1}{n^2}\text{diag}\left(X_{[1]}^\top X_{[1]}, \ldots, X_{[K]}^\top X_{[K]}\right)(v^{(t+1)} - v^{(t)}) - \frac{1}{n}X^\top\left(2w^{(t+1)} - w^{(t)}\right).$$

By rearranging the above update terms, we have

$$P_1(z^{(t)} - z^{(t+1)}) = \begin{pmatrix} \rho^{-1}I & -\frac{1}{n}X \\ -\frac{1}{n}X^\top & \frac{\rho\eta_1}{n^2}\text{diag}\left(X_{[1]}^\top X_{[1]}, \ldots, X_{[K]}^\top X_{[K]}\right) \end{pmatrix}\begin{pmatrix} w^{(t)} - w^{(t+1)} \\ v^{(t)} - v^{(t+1)} \end{pmatrix} \in \mathcal{F}(z^{(t+1)}).$$

Similarly, the updates of the distributed proximal ADMM with the second matrix choice can be equivalently written as follows:

$$w^{(t+1)} = \text{prox}_{\rho g}\left(w^{(t)} - \frac{\rho}{n} X v^{(t)}\right),$$

$$v_{[k]}^{(t+1)} = \text{prox}_{(n/\rho\eta_2)\ell_{[k]}^*}\left(v_{[k]}^{(t)} + \frac{n}{\rho\eta_2}X_{[k]}^\top\left(2w^{(t+1)} - w^{(t)}\right)\right), \quad k \in [K] \text{ (in parallel)}.$$

Using the first-order optimality conditions and rearranging terms, we have:

$$P_2\left(z^{(t)} - z^{(t+1)}\right) = \begin{pmatrix} \rho^{-1}I & -\frac{1}{n}X \\ -\frac{1}{n}X^\top & \frac{\rho\eta_2}{n^2}I \end{pmatrix}\begin{pmatrix} w^{(t)} - w^{(t+1)} \\ v^{(t)} - v^{(t+1)} \end{pmatrix} \in \mathcal{F}(z^{(t+1)}).$$

Thus, in the distributed proximal ADMM, the corresponding matrices are given by:

$$P_1 = \begin{pmatrix} \rho^{-1}I & -\frac{1}{n}X \\ -\frac{1}{n}X^\top & \frac{\rho\eta_1}{n^2}\text{diag}\left(X_{[1]}^\top X_{[1]}, \ldots, X_{[K]}^\top X_{[K]}\right) \end{pmatrix} \quad \text{and} \quad P_2 = \begin{pmatrix} \rho^{-1}I & -\frac{1}{n}X \\ -\frac{1}{n}X^\top & \frac{\rho\eta_2}{n^2}I \end{pmatrix}.$$

**Consensus ADMM.** For the standard consensus ADMM, we could equivalently write the updates in the Proposition 1 by utilizing the first-order optimality conditions as follows:

$$
\begin{aligned}
0 \quad &\in \quad \partial g(w^{(t+1)}) + \beta K \left( w^{(t+1)} - w^{(t)} + \frac{1}{n\beta K} X \left( 2v^{(t+1)} - v^{(t)} \right) \right), \\
0 \quad &\in \quad \frac{1}{n}\partial \ell^*(v^{(t+1)}) + \frac{1}{n^2\beta}\mathrm{diag}\left( X_{[1]}^\top X_{[1]}, \ldots, X_{[K]}^\top X_{[K]} \right)(v^{(t+1)} - v^{(t)}) - \frac{1}{n}X^\top w^{(t)}.
\end{aligned}
$$

By rearranging the above update terms, we have

$$
P_1(z^{(t)} - z^{(t+1)}) = \begin{pmatrix} \beta K I & \frac{1}{n}X \\ \frac{1}{n}X^\top & \frac{1}{n^2\beta}\mathrm{diag}\left( X_{[1]}^\top X_{[1]}, \ldots, X_{[K]}^\top X_{[K]} \right) \end{pmatrix} \begin{pmatrix} w^{(t)} - w^{(t+1)} \\ v^{(t)} - v^{(t+1)} \end{pmatrix} \in \mathcal{F}(z^{(t+1)}).
$$

Similarly using the first-order optimality conditions and rearranging terms, we have for the updates of the consensus ADMM with the linearization technology:

$$
P_2\left( z^{(t)} - z^{(t+1)} \right) = \begin{pmatrix} \beta K I & \frac{1}{n}X \\ \frac{1}{n}X^\top & \frac{\tau}{n^2\beta}I \end{pmatrix} \begin{pmatrix} w^{(t)} - w^{(t+1)} \\ v^{(t)} - v^{(t+1)} \end{pmatrix} \in \mathcal{F}(z^{(t+1)}).
$$

Thus, in the consensus ADMM, the corresponding matrices are given by:

$$
P_1 = \begin{pmatrix} \beta K I & \frac{1}{n}X \\ \frac{1}{n}X^\top & \frac{1}{n^2\beta}\mathrm{diag}\left( X_{[1]}^\top X_{[1]}, \ldots, X_{[K]}^\top X_{[K]} \right) \end{pmatrix} \quad \text{and} \quad P_2 = \begin{pmatrix} \beta K I & \frac{1}{n}X \\ \frac{1}{n}X^\top & \frac{\tau}{n^2\beta}I \end{pmatrix}.
$$

### C.2 Proof of Theorem 1

Let $u^{(t+1)} = P(z^{(t)} - z^{(t+1)}) \in \mathcal{F}(z^{(t+1)})$. From the convexity-concavity property of the objective function $L(w; v)$, we have that

$$
\begin{aligned}
&L(w^{(t+1)}; v) - L(w; v^{(t+1)}) \\
=&L(w^{(t+1)}; v) - L(w^{(t+1)}; v^{(t+1)}) + L(w^{(t+1)}; v^{(t+1)}) - L(w; v^{(t+1)}) \\
\leq&\langle u^{(t+1)}, z^{(t+1)} - z \rangle = \left( z^{(t)} - z^{(t+1)} \right)^\top P \left( z^{(t+1)} - z \right) \\
=&\frac{1}{2}\|z^{(t)} - z\|_P^2 - \frac{1}{2}\|z^{(t+1)} - z\|_P^2 - \frac{1}{2}\|z^{(t)} - z^{(t+1)}\|_P^2 \\
\leq&\frac{1}{2}\|z^{(t)} - z\|_P^2 - \frac{1}{2}\|z^{(t+1)} - z\|_P^2,
\end{aligned}
$$

where the last inequality follows from the fact that $\|\cdot\|_P$ is a semi-norm. Thus, we have

$$
L(\bar{w}^{(T)}; v) - L(w; \bar{v}^{(T)}) \leq \frac{1}{T}\sum_{t=0}^{T-1}\left\{ L(w^{(t+1)}; v) - L(w; v^{(t+1)}) \right\} \leq \frac{1}{2T}\|z^{(0)} - z\|_P^2,
$$

where the first inequality comes from the convexity-concavity of $L(w; v)$ and the second inequality comes from the above relation.

### C.3 Proof of Theorem 2

We first show that $\{z^{(t)}\}$ is bounded. Let $z^* = (w^*, v^*)$ denote the optimal solution of the saddle point problem equation SP. Firstly, from the convexity-concavity property of the objective function $L(w; v)$, we

have that

$$
\begin{aligned}
&L(w^{(t+1)}; v) - L(w; v^{(t+1)}) \\
&= L(w^{(t+1)}; v) - L(w^{(t+1)}; v^{(t+1)}) + L(w^{(t+1)}; v^{(t+1)}) - L(w; v^{(t+1)}) \\
&\leq \langle u^{(t+1)}, z^{(t+1)} - z \rangle = \left(z^{(t)} - z^{(t+1)}\right)^{\top} P \left(z^{(t+1)} - z\right) + \langle \epsilon^{(t+1)}, z^{(t+1)} - z \rangle \\
&= \frac{1}{2}\|z^{(t)} - z\|_P^2 - \frac{1}{2}\|z^{(t+1)} - z\|_P^2 - \frac{1}{2}\|z^{(t)} - z^{(t+1)}\|_P^2 + \langle \epsilon^{(t+1)}, z^{(t+1)} - z \rangle \\
&\leq \frac{1}{2}\|z^{(t)} - z\|_P^2 - \frac{1}{2}\|z^{(t+1)} - z\|_P^2 + \langle \epsilon^{(t+1)}, z^{(t+1)} - z \rangle \\
&\leq \frac{1}{2}\|z^{(t)} - z\|_P^2 - \frac{1}{2}\|z^{(t+1)} - z\|_P^2 + \|\epsilon^{(t+1)}\| \|z^{(t+1)} - z\|.
\end{aligned}
\tag{9}
$$

Choosing $z = z^*$ and using $L(w^{(t+1)}; v^*) - L(w^*; v^{(t+1)}) \geq 0$, we have

$$
\begin{aligned}
\frac{1}{2}\|z^{(t+1)} - z^*\|_P^2 &\leq \frac{1}{2}\|z^{(t)} - z^*\|_P^2 + \|\epsilon^{(t+1)}\| \|z^{(t+1)} - z^*\| \\
&\leq \frac{1}{2}\|z^{(t)} - z^*\|_P^2 + C\|\epsilon^{(t+1)}\| \|z^{(t+1)} - z^*\|_P,
\end{aligned}
$$

where $C$ is a constant satisfying $\|z\| \leq C\|z\|_P$, which exists since $P$ is positive definite. Therefore,

$$
(\|z^{(t+1)} - z^*\|_P - C\epsilon^{(t+1)})^2 \leq \|z^{(t)} - z^*\|_P^2 + (\epsilon^{(t+1)})^2.
$$

Taking square root of both sides yields that

$$
\begin{aligned}
\|z^{(t+1)} - z^*\|_P - C\epsilon^{(t+1)} &\leq \sqrt{\|z^{(t)} - z^*\|_P^2 + (\epsilon^{(t+1)})^2} \\
&\leq \|z^{(t)} - z^*\|_P + \epsilon^{(t+1)}.
\end{aligned}
$$

Simple induction gives that

$$
\|z^{(T)} - z^*\|_P \leq \|z^{(0)} - z^*\|_P + (C+1) \sum_{t=1}^{T} \epsilon^{(t)}
$$

As a result, we have

$$
\sup_T \|z^{(T)} - z^*\| \leq C \sup_T \|z^{(T)} - z^*\|_P \leq C\|z^{(0)} - z^*\|_P + C(C+1) \sum_{t=1}^{\infty} \epsilon^{(t)},
$$

which gives the boundedness of $\{z^{(t)}\}$.

Let $u^{(t+1)} \in \mathcal{F}(z^{(t+1)})$. From equation 9, we have that

$$
\begin{aligned}
L(w^{(t+1)}; v^*) - L(w^*; v^{(t+1)}) &\leq \frac{1}{2}\|z^{(t)} - z^*\|_P^2 - \frac{1}{2}\|z^{(t+1)} - z^*\|_P^2 + \|\epsilon^{(t+1)}\| \|z^{(t+1)} - z^*\| \\
&\leq \frac{1}{2}\|z^{(t)} - z^*\|_P^2 - \frac{1}{2}\|z^{(t+1)} - z^*\|_P^2 + D\|\epsilon^{(t+1)}\|,
\end{aligned}
$$

where the last-second inequality follows from Cauchy-Schwarz inequality and the last inequality from the definition of $D$. Thus, we have

$$
\begin{aligned}
L(\bar{w}^{(T)}; v^*) - L(w^*; \bar{v}^{(T)}) &\leq \frac{1}{T} \sum_{t=0}^{T-1} \left\{ L(w^{(t+1)}; v^*) - L(w^*; v^{(t+1)}) \right\} \\
&\leq \frac{1}{2T}\|z^{(0)} - z^*\|_P^2 + \frac{D \sum_{t=1}^{T} \|\epsilon^{(t)}\|}{T},
\end{aligned}
\tag{10}
$$

where the first inequality comes from the convexity-concavity of $L(w; v)$ and the second inequality comes from the above relation.

# D  Experimental Details

## D.1  Experiment 2 data generation details

We generate $n = 3000$ training examples $\{x_i, y_i\}_{i=1}^n$ according to the model $y_i = \langle x_i, w^* \rangle + \epsilon_i, \epsilon_i \sim \mathcal{N}(0,1)$, where $x_i \in \mathbb{R}^d$ with $d = 500$. The samples are distributed uniformly on $K = 30$ machines. We set $w^*$ as the vector of all ones, whereas for the $\ell_1$ penalty, we let the first 100 elements of $w^*$ be ones and the rest be zeros. For the generation of $x_i$'s, we designed two cases: IID data and non-IID data. (1) Under the IID setting, we generate each $x_i \sim \mathcal{N}(0, \Sigma)$ where the covariance matrix $\Sigma$ is diagonal with $\Sigma_{j,j} = j^{-2}$. This covariance setting renders an ill-conditioned dataset, making it a challenging situation of solving distributed and large-scale optimization problem. (2) To generate the non-IID case, we follow a setup similar to the one in Zhou & Li (2023). Specifically, we generate $[n/3]$ samples $x_i$ from the standard normal distribution, $[n/3]$ samples from the Student's $t$-distribution with 5 degrees of freedom, and the rest samples are from the uniform distribution on $[-5, 5]$. After generating all the samples, we shuffle them and randomly distribute them across $K$ machines.

## D.2  Experiment 3: Binary Classification with Real Data

Finally, we test the performance of the five update rules on regularized SVM classification problem using real datasets.

**Datasets.**  The real datasets from the LibSVM library Chang & Lin (2011) used in the study are `a1a`, `w8a`, and `real-sim`. Details of each dataset, including the number of samples and features are summarized in Table 1. In our experiments, all samples in each dataset are evenly distributed on the machines. We select different numbers of machines for each dataset to evaluate the performance of the proposed approach, where the number of machines is also given in Table 1.

**Method.**  We use the update rules to solve two regularized SVM classification problems:

$$\min_{w \in \mathbb{R}^d} \frac{1}{n} \sum_{k=1}^{K} \sum_{i \in \mathcal{P}_k} \max\left(0, 1 - y_i w^\top x_i\right) + g(w),$$

where the penalty function $g(w)$ are selected as either (1) $\ell_1$ penalty $\lambda\|w\|_1$ and (2) the $\ell_2$ penalty $\frac{\lambda}{2}\|w\|^2$. We use three real datasets in LibSVM package for each problem. We choose the regularization parameter $\lambda = \frac{1}{n}$ in all experiments conducted in this subsection. Like Experiment 2, we select the optimal parameters for $\beta$ and $\rho$ and prescribe the values for all other tuning parameters.

**Results.**  The results are shown in Figure 4. They again validated that the performance of Consensus-PD and Proximal-1-PD are almost overlapping, and LinConsensus-PD and Proximal-2-PD are almost overlapping. All ADMM variants consistently outperform the CoCoA method across all experiments. Finally, Consensus-PD and Proximal-1-PD achieve the better performance compared with LinConsensus-PD and Proximal-2-PD. These results confirms with the study in Experiment 2. However, in the SVM with lasso penalty scenarios, LinConsensus-PD and Proximal-2-PD exhibit slightly less stability compared to Consensus-PD and Proximal-1-PD.

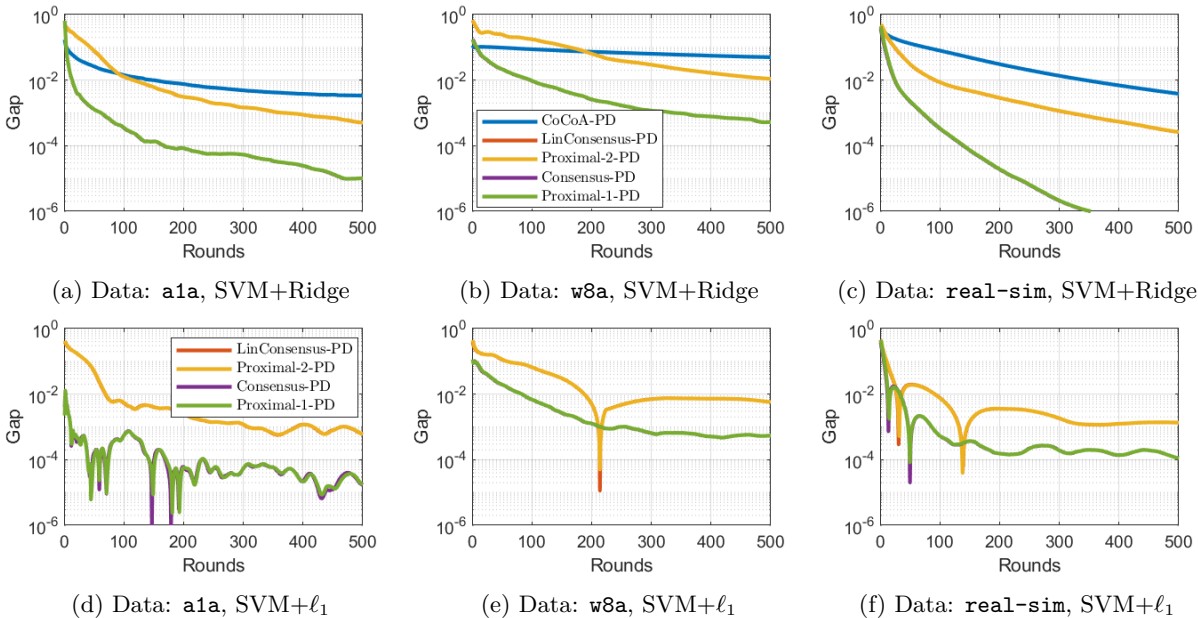

Figure 4: Relative gap differences versus the number of communication rounds for various real datasets across different models. The first row of plots illustrates the results for SVM with a ridge penalty across different datasets, while the second row shows the results for SVM with a lasso penalty across the same datasets.

