# OpenReview forum: "On the Relationship Between CoCoA and ADMM for Distributed Empirical Risk Minimization"
_TMLR — Accepted by TMLR_

### Review · Reviewer_D7wh · 2026-03-20

**Summary Of Contributions:**

The authors present an interesting and elegant conceptual framework that seeks to unify several prominent distributed optimization algorithms under a single primal-dual update structure namely CoCoA, Consensus ADMM, Linearized Consensus ADMM, and two Proximal ADMM variants. The paper demonstrates that under specific parameter settings (in particular with ridge regularization), the dual updates of these methods can be aligned. Furthermore, the authors leverage this unified form to provide a common O(1/T) ergodic convergence analysis and offer empirical studies on parameter sensitivity across synthetic and real-world datasets.

Strengths:
1, The core idea of bridging the CoCoA and ADMM literatures through a unified primal-dual lens is highly valuable and provides a nice theoretical synthesis. The side-by-side presentation of Equations (1) to (5) makes the structural similarities immediately visible.
2, The parameter sensitivity studies (Experiment 1 & Figure 2) are practically useful for researchers implementing these distributed methods.
3, Showing that Consensus ADMM on the min-max problem is equivalent to Proximal ADMM on the max-min problem is a clean and instructive observation.

Weaknesses:

1, Several mathematical claims, including the title's assertion and the exactness of the algorithmic equivalence, appear overstated compared to what the proofs actually support.
2, There are technical gaps in the convergence analysis, specifically regarding matrix definiteness assumptions, an incomplete proof step in Proposition 1, and the translation of theoretical inexactness to practical solver behavior.
3, The convergence theory is limited to known sublinear rates and does not exploit the strong convexity structure that is central to the very setting (ridge-regularized ERM) where the main equivalence result holds.

**Audience:**

Yes

**Audience Explanation:**

The TMLR audience includes many researchers working on distributed optimization, federated learning, and operator-splitting methods. Bridging the communication-efficient ERM community with CoCoA and the ADMM community is a relevant and interesting pursuit. The unified update rules and the resulting ability to easily compute the duality gap will be of practical interest.

**Broader Impact Concerns:**

No.

**Claims And Evidence:**

No

**Claims Explanation:**

I appreciate the unified perspective offered by the paper. Bridging the communication-efficient ERM literature around CoCoA and the ADMM literature is a relevant and interesting direction. However, I have several major mathematical and foundational concerns that should be addressed, because some of the current claims go beyond what is actually established by the proofs.

1. The extent of the equivalence claim is overstated. The title and abstract suggest a broad algorithmic equivalence between CoCoA and ADMM. In contrast, Corollary 1 only establishes coincidence of the dual iterates $v^{(t)}$, and only in the ridge-regularized case $g(w)=\lambda ||w||_2^2/2$ under specific parameter alignment. In particular, the surrounding discussion makes clear that one needs $\rho=\lambda^{-1}$ and $\eta_1=\sigma$, and under the recommended CoCoA choice $\gamma=1$ and $\sigma=K$, this becomes $\eta_1=K$. The manuscript also explicitly notes that the $w$-updates are different, since Proximal-1-PD produces an exponentially weighted average of the CoCoA w-updates. Therefore, claiming that the two methods are simply the same algorithm is stronger than what is actually shown.

2. There is a mismatch between the assumption $P \succ 0$ in Theorem 2 and the parameter choices justified earlier in the paper. The proof of Theorem 2 uses the existence of a constant C such that $||z|| \leq C ||z||_P$，which requires P to be positive definite. However, Lemma 1 and the recommended choices such as $\eta_1=K$ only guarantee positive semidefiniteness. If P has a nontrivial null space, which can happen when $X[k]^\top X[k]$ is singular, for example when $d>n_k$, then this norm comparison fails. In that case, the boundedness argument for $\{z^{(t)}\}$ and the subsequent convergence steps are not justified as written.

3. The discussion after Theorem 1 conflates ergodic gap convergence with convergence of the iterates. Theorem 1 gives an $O(1/T)$ bound on the ergodic saddle quantity $ L(\bar{w}^{(T)};v)-L(w;\bar{v}^{(T)}) $, and in particular on $ L(\bar{w}^{(T)};v^\ast)-L(w^\ast;\bar{v}^{(T)}) $.
Without additional assumptions such as strong convexity, strict concavity, or uniqueness of the saddle point, this does not by itself imply that $\bar{z}^{(T)}\to z^\ast$ at rate $O(1/T)$. The current wording is therefore too strong.

4. There is a logical gap in Proposition 1 and Appendix B.1. The proof introduces $\tilde v[k]$ as the dual variable of the local $w_k$-subproblem and then derives updates in terms of $\tilde v[k]$. The manuscript then states that one still needs to show that $\tilde v$ converges to the global dual variable v, and that this will be addressed later. However, this step is not explicitly completed. Since the claimed primal-dual reformulation of consensus ADMM depends on identifying these variables correctly, this missing argument is not a minor detail.

5. The inexactness assumption in Theorem 2 is not well aligned with the practical usage emphasized by the paper. The theorem assumes $ \sum_{t=1}^{\infty} ||\epsilon^{(t)}||_2 < \infty $, which means that the inner errors must be summable across rounds. In practice, however, a major appeal of distributed ERM methods such as CoCoA is precisely that one often uses a fixed amount of local work per communication round. The manuscript should explain more clearly how the theoretical assumption relates to the practical regime studied in the experiments.

**Requested Changes:**

1. Revise the main equivalence claims in the title, abstract, and introduction so that they accurately match the proven result. The paper should state clearly that the connection to CoCoA is established for CoCoA-PD in the ridge-regularized case $g(w)=\lambda ||w||_2^2/2$, under specific parameter alignment, and that the equivalence is at the level of the dual iterates rather than the full primal-dual trajectory.

2. Resolve the $P \succ 0$ issue in Theorem 2. Either give explicit conditions under which the matrices $P_1,\dots,P_4$ are positive definite, or revise the proof so that it works under positive semidefiniteness. If additional assumptions on the data matrices X[k] are needed, they should be stated clearly.

3. Close the logical gap in Proposition 1 by providing the missing argument that justifies the passage from the local dual variables $\tilde v[k]$ to the global dual blocks v[k]. As written, this identification is asserted but not fully proved.

4. Rewrite the interpretation of Theorem 1 so that it states precisely what is proved. The theorem establishes ergodic primal-dual gap control. Any claim that $\bar{z}^{(T)}$ converges to $z^\ast$ should either be removed or supported by explicit additional assumptions.

5. It would be nice to add a brief discussion in Section 5 clarifying how the summability condition $ \sum_{t=1}^{\infty} ||\epsilon^{(t)}||_2 < \infty $ relates to the common practical regime in which each machine performs a fixed amount of local computation per round.

6. Could you please expand the derivation in Appendix B.5? When Consensus-PD is applied to the saddle formulation with negated objective $-L(w;v)$, it would help to write out a few more intermediate algebraic steps showing how the sign changes, proximal terms, and parameter mapping $ \rho^{-1}=\beta K, \eta_1=K $ lead to the Proximal-1-PD update in equation (3). This would make Corollary 2 substantially easier to verify.

---

> ### Author Response · Authors · 2026-05-03
> **Point-to-point answers to the detailed suggestions**
>
> We sincerely thank the reviewer for the constructive and detailed feedback, which has significantly improved the clarity and rigor of the manuscript. We have updated the manuscript accordingly, with all changes highlighted in red.
>
> > *Overstated equivalence claim*
>
> **Response:** We agree with the reviewer on the scope of the equivalence, and followed the reviewer's suggestion to revise the title, abstract and introduction to clarify the scope of the equivalence results. We revised the title and provided detailed scope in the significantly-revised introduction.
>
> >*Positive definite P in Theorem 1, 2*
>
> **Response:** There is indeed a mismatch between Theorem 1, 2 and the proximal ADMM introduced in Section 3.2. We added Remark 1, following Theorem 2, to explicitly state sufficient conditions under which each of the four matrices P1 - P4 are strictly positive definite, by using the Schur Complement rule. Due to space limit, please refer to the revised article for the detailed conditions and discussions.
>
> >*Logical gap in Proposition 1 on passage from the local dual variables  to the global dual blocks*
>
> **Response:** We thank the reviewer for identifying this gap. In the revised Appendix B.1, we have expanded the proof of Proposition 1 to explicitly justify the identification of $\tilde{v}{[k]}$ , with the global dual block $v_{[k]}$. Specifically, we now show that after eliminating the auxiliary primal variables $w_k$ and dual multipliers $u_k$ via the KKT conditions of the local subproblems, the local Lagrange multiplier $\tilde{v}{[k]}$  introduced in the dualization of the $w_k$-update coincides with the corresponding block of the global dual variable $v$. The key step is to verify that $u_k^{(t+1)} = \frac{1}{n} X_{[k]}\tilde{v}{[k]} ^{(t+1)}$, which links the consensus ADMM multipliers to the dual variables in the saddle-point formulation. We then invoke Lemma 2 and Theorems 1–2 to conclude that $\tilde{v}^{(t)}$ converges to $v^*$, completing the argument.
>
> >*Rewrite the interpretation of Theorem 1.*
>
> **Response:** We agree that the original interpretation overstated the result. We have revised the discussion following Theorem 1 to read:
> > *"Under the assumption that objective being strongly convex-concave or that the saddle point being unique, $\bar{z}^{(T)}$ converges to the optimal solution at a rate of $O(1/T)$."*
> This makes it explicit that convergence of the iterates themselves requires additional conditions beyond what Theorem 1 alone provides.
>
> >*Clarifying summable condition in Theorem 2*
>
> **Response:** We thank the reviewer for the inquiry. The term $\epsilon^{(t)}$ represents the sub-gradient residual of the $v$-subproblem at iteration $t$ $v$-step update,
> $$v_{[k]}^{(t+1)}\gets \argmin_{v_{[k]}} \frac 1 n \sum_{i\in P_k} \ell_i^*(v_i) - \frac 1 n \langle v_{[k]}, X_{[k]}^{\top} w^{t}\rangle + \frac 1 2 \Vert  v_{[k]}- v_{[k]}^{(t)}\Vert_M, $$
> where $M$ is bottom-right block of $P$, i.e., $c_1 I$ or $c_2 B$ in the article, where $c_1$ and $c_2$ are constants.
>
> One approach of specifying the number of iterations is to ensure $\Vert\epsilon^{(t)}\Vert$ is below a limit, for example $1/t^2$. If a standard inner solver such as Gradient Descent exhibits linear convergence: $\|\epsilon^{(t, j)}\|_2 \leq C \cdot \rho^j$, the number of inner iterations $j$ must be increased logarithmically. However, for some losses such as logistic loss, the conjugate is not globally smooth, higher number of inner iterations may be needed. In practice, however, it is unnecessary to increase the number of inner iterations indefinitely, as the sub-gradient error eventually reaches the  machine precision. At this stage, further inner iterations do not yield meaningful improvements.
>
> We incorporated the above discussion in the revision.
>
> >*Details in proof of Corollary 2*
>
> We expanded the proof of Corollary 2 in Appendix B.5. The revised derivation now includes the following additional details:
>
> (a) We write out the Consensus-PD updates applied to the negated objective $-L(w; v)$, clearly showing how the sign change swaps the roles of the $\arg\max$ and $\arg\min$ steps.
>
> (b) We introduce the shifted primal sequence $\tilde{w}^{(t)} := w^{(t+1)}$ and show step by step how the extrapolation $2w^{(t+1)} - w^{(t)}$ transforms under this relabeling.
>
> (c) We carry out the parameter substitution $s_1 = \frac{1}{n^2\beta}$, $s_2 = \beta K$, $M_1 = \mathrm{diag}(X_{[1]}^\top X_{[1]}, ... , X_{[K]}^\top X_{[K]})$, $M_2 = I$, and explicitly show that the resulting $v$-update and $w$-update match the Proximal-1-PD form in Equation (3) when $\eta_1 = K$ and $\rho = \frac{1}{\beta K}$.
>
> We believe the expanded derivation now makes Corollary 2 substantially easier to verify.

---

> > ### Comment · Reviewer_D7wh · 2026-05-05
> >
> > Thanks for the reply and modifications. I have no further questions.

---

### Review · Reviewer_MzUs · 2026-04-05

**Summary Of Contributions:**

This paper considers primal-dual methods for distributed ERMs, and focuses on two method classes: communication-efficient distributed dual coordinate ascent (CoCoA) and consensus + proximal + linearized ADMM. All above methods can be in a form with only primal and dual variables: CoCoa is proximal ADMM for dual problem; consensus ADMM is proximal ADMM. Then this paper observes adjusting AL parameters can make ADMM variants outperform CoCoA variants. Then, linear ADMM and tunings are discussed. Simulations and data analysis support previous theories.

**Audience:**

Yes

**Audience Explanation:**

Distributed CoCoA & ADMM could be of interest to people.

**Claims And Evidence:**

Yes

**Claims Explanation:**

Solid proofs to theoretical claims, and sufficient numerical experiments.

**Requested Changes:**

Claimed contributions in this paper are well-proven. However, one can question whether contributions in this paper are large enough. For example, after much analysis, this paper 1) didn't propose any specific novel algorithm, that is "optimal" in some sense according to those previous analysis, and to be proven to be 2) theoretically near-optimal + 3) numerically efficient.
The decision is up to AE on whether the above change is mandatory for TMLR submission.

---

> ### Author Response · Authors · 2026-05-03
> **Appreciation and summary of contributions**
>
> We thank the reviewer for the positive assessment of the paper, especially for recognizing the solidity of the theoretical proofs and the sufficiency of the numerical experiments. We also appreciate the reviewer’s view that the topic should be of interest to the TMLR audience.
>
> We understand the reviewer’s concern that the paper does not propose a specific new algorithm that is theoretically near-optimal and numerically superior. We agree that this paper is not primarily an algorithm-design paper. Its main contribution is instead a new structural understanding of existing distributed ERM methods through a unified primal-dual framework.
>
> In our view, this contribution is substantial.
> 1. The paper shows that CoCoA-type methods and ADMM-type methods are not fundamentally separate families: for example, CoCoA with $\ell_2$ ridge regularization is a special case of consensus ADMM with the same dual update.
>
> 2. The paper also shows that consensus ADMM on the primal problem is equivalent to proximal ADMM on the dual problem. These observations provide a deeper understanding of the algorithmic landscape and explain, in a principled way, how the augmented-Lagrangian / dual-update parameters control performance. This is precisely why tuned ADMM variants can outperform CoCoA variants.
>
> 3. Moreover, the framework leads to practical consequences, not only conceptual ones. It yields a natural primal-dual gap stopping criterion for consensus ADMM, clarifies how to tune the relevant parameters, and provides guidance that is supported by our experiments. We therefore believe the paper makes a meaningful contribution even without introducing a brand-new optimizer.
>
> That said, we agree with the reviewer that designing a new provably near-optimal and numerically efficient algorithm within this framework would be an exciting next step. In the revision, we will clarify more explicitly that the goal of the present paper is unification, clarification, and parameter understanding, rather than proposing a new optimal method, and we will state this future direction more clearly.
>
> We sincerely thank the reviewer for this helpful suggestion.

---

### Review · Reviewer_fgeQ · 2026-04-19

**Summary Of Contributions:**

The paper aims to unify several distributed optimization algorithms under one primal-dual framework, similar in spirit to the linearized proximal ADMM method. The unificiation allows all of these algorithms to be analyzed within one framework to provide a rigorous of the effect of the algorithm's dual update hyperparameters on its performance.

**Audience:**

Yes

**Audience Explanation:**

Putting a unifying lens on a diverse family of algorithms used in machine learning facilitates new perspectives, insights, and perhaps better scientific discoveries.

**Claims And Evidence:**

No

**Claims Explanation:**

I don't know how true it is, but it appears this work stems from the siloed machine learning community that is not aware that such a unification was already done, and the main contribution of this paper is showing that CoCoA with a ridge penalty is also a member of the "linearized proximal ADMM" family.

The deterministic optimization community has been doing research in ADMM, and other primal-dual splitting algorithms for a very long time, and has unified many algorithms as instances of Linearized Proximal ADMM. For an example, see the 2011 work by Esse et. al., [3], 2014 work by Sheffi and Teboulle [1], references therein, and Chapter 15 in the First Order Methods book by Beck [2].

Thus, I believe the claims of this paper should be significantly revisited and made much more precise, given the already existing body of work in this area. Consequently, this paper requires significant work before it is ready for publication.

---
**References**

[1]: Shefi, Ron, and Marc Teboulle. "Rate of convergence analysis of decomposition methods based on the proximal method of multipliers for convex minimization." SIAM Journal on Optimization 24, no. 1 (2014): 269-297.

[2]: Beck, A. (2017). First-order methods in optimization. Society for Industrial and Applied Mathematics.

[3]: Esser, E., Zhang, X., & Chan, T. F. (2010). A general framework for a class of first order primal-dual algorithms for convex optimization in imaging science. SIAM Journal on Imaging Sciences, 3(4), 1015-1046.

**Requested Changes:**

Revise the paper in line with the existing body of work on the unification of many algorithms under the proximal linearized alternating minimization lens, and make the contributions of this paper more precise in line with this body of work. Do not consider papers only from ML conferences - there is a very large body of work in the classical optimization community.

---

> ### Author Response · Authors · 2026-05-03
> **Originality of the results and comparison with the suggested references**
>
> We thank the reviewer for the comments. Yet respectfully disagree with the comment by the reviewer, that "such a unification was already done".
>
> The reviewer said "(optimization community) has unified many algorithms as instances of Linearized Proximal ADMM", yet in the listed literature, these works focus on the relationship between standard two blocks ADMM/Proximal ADMM algorithms with other primal-dual setups: primal-dual hybrid gradients (PDHG), proximal gradient descent (PGD), alternating minimization algorithm (ADA), Douglas-Rachford splitting, and so on.
>
> These existing work does not cover the discovered associations between CoCoA, consensus ADMM, and proximal ADMM at all. (1) None of the existing work listed by the reviewer involves any distributed optimization/federated learning algorithms, including consensus ADMM or CoCoA. (2) The splitting mechanism of CoCoA is based on coordinate ascent of the dual problem, but based on primal-dual updates.
>
> Here is a detailed comparison between the scope of the suggested literature and our work:
>
> - In Esser, E., Zhang, X., & Chan, T. F. (2010):
>
>   - This paper aims to unify PDHG, split inexact Uzawa, proximal gradient descent (termed proximal forward-backward splitting (PFBS)), alternating minimization algorithm (AMA), two-block ADMM, and Douglas-Rachford Splitting. All solve the problem types $\min_x f(Ax) + g(x)$, where $f$ and $g$ are convex functions (including indicator functions).
>
>   - They do not cover the equivalence between proximal ADMM (replace customized norm $P$ in proximal updates of ADMM algorithm), consensus ADMM (use duplicated variables to minimize $\sum f_i(x)$), and CoCoA (use splittable coordinate ascent of the dual problem). Moreover, the three algorithms we discribed are all aimed for distributed optimization (federated learning).
>
>   - The only similarity between our paper and this paper is that we used overlapping suite of mathematical techniques in deriving the equivalence between methods: (1) investigating the dual problem (section 2.1), (2) spliting the primal problem with additional constraint $w=Au$,  and (3) utilizing the Moreau  decomposition.
>
> - In Shefi, Ron, and Marc Teboulle. (2014):
>   - This paper interprets how Proximal ADMM (termed as AD-PMM in that paper) and PDHG (termed as Chambolle-Pock algorithm) can be originate from proximal method of multiplier (PMM). They focus on two-block example.
>
>   - Again, they do not link Proximal ADMM with specific choices of $Q_1$ and $Q_2$ with *consensus* ADMM or CoCoA. The paper does not consider distributed optimization/federated learning setup.
>
> - In Beck (2017):
>   - This book described primal/dual projected subgradient methods, mirror descent, proximal gradient descent (with Block Proximal Gradient and dual-based algorithms), conditional gradient methods, which are not the focus of our paper.
>   - Though it involved coordinate descent algorithm in Chapter 14, but it did not cover how it can be applied on the dual to facilitate parallization (CoCoA). Though it involved ADMM and proximal ADMM (termed AD-PMM) in Chapter 15, it did not introduce the distributed setup, and not introduced the connection between proximal ADMM, *consensus* ADMM and CoCoA.
>
>
> The reviewer suggested "the claims of this paper should be significantly revisited and made much more precise" and  "revise the paper in line with the existing body of work on the unification of many algorithms under the proximal linearized alternating minimization lens". We agree that we should state our scope clearer in the introduction and refer to the existing body of work.
>
> For this:
>
> - We have expanded the introduction, clarifying the scope of the connections between algorithms, in the paragraph that starts with "In this paper, we will show an interesting connection between CoCoA, consensus ADMM, and distributed proximal ADMM through a primal-dual reformulation."
>
> - We described how the existing work are different from the unification of the primal-dual algorithms that the reviewer pointed out: "Besides, the discovery is original from the extensive existing studies on the connection between primal-dual algorithms (Esser et al., 2010; Shefi & Teboulle, 2014; Beck, 2017) in the past few decades, which focused on associating two-block ADMM and its proximal variant with other algorithms such as Primal-Dual Hybrid Gradient (PDHG), Proximal Gradient Descent, Alternating Minimization, and Douglas-Rachford Splitting".
>
> - As pointed out by another reviewer, the conclusion is only applicable for ridge penalty, not applicable for other penalties. We agree with that comment and carefully addressed this issue in article.

---

> > ### Comment · Reviewer_fgeQ · 2026-05-16
> > **Reply**
> >
> > Dear authors.
> >
> > To the best of my belief, the correct framing of this paper is "we use an **existing and well known** mathematical technique, of unifying families of algorithms as linearized proximal ADMM with a pair of distance matrices to make **new connections**".
> >
> > But instead, a cosmetic change has been introduced to just cite previous work. I think that without such framing, the paper "oversells".

---

> > > ### Author Response · Authors · 2026-05-22
> > > **Reply**
> > >
> > > Thanks for the reply of the reviewer.
> > >
> > > After the first revision,  the introduction was significantly reframed and only focused on our results:
> > >
> > > - Para 1: the ERM problem of interest,
> > > - Para 2: the two classes of algorithms CoCoA and ADMM family
> > > - Para 3: summarized the approach,
> > > - Para 4: show the main results
> > > - Para 5: give consequences
> > >
> > > With this additional comments, we further changed Para 3:
> > >
> > > - Old: In this paper, we will show an interesting connection between CoCoA with ridge regularization, consensus ADMM, and distributed proximal ADMM through a primal-dual reformulation. This connection is not obvious, because these algorithms are based on distinct paradigms of parallization: CoCoA on separable dual coordinate ascent, consensus ADMM on a constrained primal reformulation with duplicated variables, and proximal ADMM originated in a penalized twoblock ADMM format.
> > >
> > > - New: In this paper, we will show an interesting connection between CoCoA with ridge regularization, consensus ADMM, and distributed proximal ADMM through a primal-dual reformulation. **The technique is to reformulate these algorithms into iterative primal and block-dual update, and then unify them as proximal ADMM with a pair of distance matrices. Though the expressiveness of proximal ADMM has been revealed where a few algorithms are unified with similar technique (see
> > > Section 1.1 of Deng & Yin (2016)), the algorithms involved in our paper were not covered.** Moreover, their unification is not obvious, given that the parallelization are based on distinct paradigms: CoCoA on separable dual coordinate ascent, consensus ADMM on a constrained primal reformulation with duplicated variables, and proximal ADMM originated in a penalized two-block ADMM format.
> > >
> > > The reviewer may checked the new version in updated submission.

---

> > > > ### Comment · Action_Editor_zZuh · 2026-05-22
> > > > **Update title**
> > > >
> > > > Dear authors,
> > > >
> > > > Thanks for the revisions.  Could you please change the metadata for the submission so that the new title is updated (to "On the Relationship Between CoCoA and ADMM for Distributed Empirical Risk Minimization")?
> > > >
> > > > Thanks
> > > > -Stephen

---

> > > > > ### Author Response · Authors · 2026-05-22
> > > > > **Title updated**
> > > > >
> > > > > Thanks, Stephen. This has been done.

---

### Decision · Action_Editor_zZuh · 2026-05-27

**Recommendation:** Accept as is

**Audience:**

Yes

**Audience Explanation:**

Some of the reviewers had concerns about the novelty of the connection, feeling that (at least at a high level) some of this was already known.  The revisions have mostly addressed this: the authors are now more specific (and have updated the title), and give more credit to the literature. The reviewers do think there is enough novelty to make this interesting (and the overall topic, ADMM-style methods, is very popular)

**Claims And Evidence:**

Yes

**Claims Explanation:**

The derivations and math about the relationship are solid, and no reviewers questioned them.